# LEARNING MULTIPLEX EMBEDDINGS ON TEXT-RICH NETWORKS WITH ONE TEXT ENCODER

## ABSTRACT

In real-world scenarios, texts in a network are often linked by multiple semantic relations (*e.g.,* papers in an academic network are referenced by other publications, written by the same author, or published in the same venue), where text documents and their relations form a *multiplex text-rich network*. Mainstream text representation learning methods use pretrained language models (PLMs) to generate one embedding for each text unit, expecting that all types of relations between texts can be captured by these single-view embeddings. However, this presumption does not hold particularly in multiplex text-rich networks. Along another line of work, multiplex graph neural networks (GNNs) directly initialize node attributes as a feature vector for node representation learning, but they cannot fully capture the semantics of the nodes' associated texts. To bridge these gaps, we propose METERN, a new framework for *learning Multiplex Embeddings on TExt-Rich Networks*. In contrast to existing methods, METERN uses one text encoder to model the shared knowledge across relations and leverages a small number of parameters per relation to derive relation-specific representations. This allows the encoder to effectively capture the multiplex structures in the network while also preserving parameter efficiency. We conduct experiments on nine downstream tasks in five networks from both academic and e-commerce domains, where ME-TERN outperforms baselines significantly and consistently. Code is available at `https://anonymous.4open.science/r/METERN-ICLR24-C6CD`.

## 1 INTRODUCTION

Texts in the real world are often interconnected by multiple types of semantic relations. For example, papers connected through the "`same-venue`" relation edges lean towards sharing coarse-grained topics, while papers connected through the "`cited-by`" relation edges tend to share fine-grained topics; e-commerce products linked by the "`co-viewed`" edges usually have related functions, while products linked by the "`co-brand`" edges can have similar designs. The texts and multiple types of links together form a type of networks called *multiplex text-rich networks*, where documents are treated as nodes and the edges reflect multiplex relations among documents. Given a multiplex text-rich network, it is appealing to learn node representations tailored for different relation types, which can be broadly utilized in various downstream tasks (*e.g.,* paper classification and recommendation in academic networks, and item recommendation and price prediction in e-commerce networks).

A straightforward way to learn node representations in a text-rich network is to encode the nodes' associated texts with pretrained language models (PLMs) (Brown et al., 2020; Devlin et al., 2019; Liu et al., 2019b). Many studies (Cohan et al., 2020; Ostendorff et al., 2022; Reimers & Gurevych, 2019) propose to finetune PLMs with contrastive objectives, pushing the text representations with similar semantics to be close in the latent space while pulling those unrelated apart. The semantics correlation between text units is then measured by the similarity of their representations, such as cosine similarity or dot product. However, most existing approaches based on PLMs use a single vector for each text unit, with the implicit underlying assumption that the semantics of different relations between text units are largely analogous, which does not hold universally, particularly in multiplex text-rich networks (demonstrated in Section 3).

To capture the diverse relationships among nodes, multiplex representation learning (Jing et al., 2021; Park et al., 2020; Qu et al., 2017; Zhang et al., 2018) is proposed in the graph domain. The philosophy is to learn multiple representations for every node, each encoding the semantics of one relation. They mainly adopt graph neural networks (GNNs) as the backbone model architecture and utilize *separate encoders* to encode each relation. Nevertheless, these studies with GNNs represent the texts associated with each node as bag-of-words or context-free embeddings (Mikolov et al., 2013), which are not enough to characterize the contextualized text semantics. For example, "*llama*" in biology

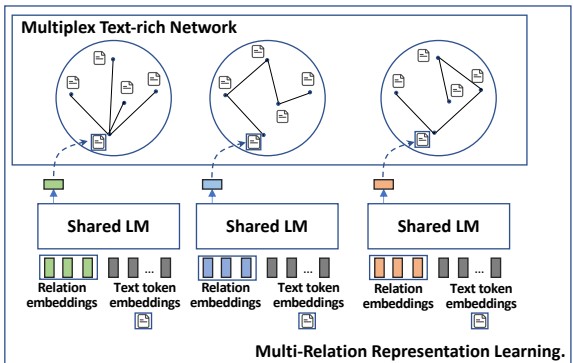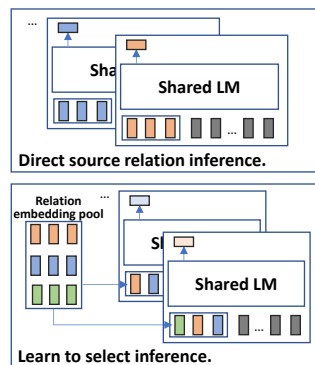

Figure 1: Model Framework Overview. METERN has a language model encoder to model the shared knowledge among relations and relation-prior embeddings to capture the relation-specific signals.

books and "*llama*" in machine learning papers should have different meanings given their context, but they correspond to the identical entry in a bag-of-words vector and possess the same context-free embedding. To capture contextualized semantics, a straightforward idea is to adopt PLMs (Brown et al., 2020; Devlin et al., 2019; Liu et al., 2019b). However, it is inefficient and unscalable to have separate PLMs for each relation, given that one PLM usually has millions or billions of parameters (Kaplan et al., 2020).

To this end, we propose METERN to learn multiplex node/text embeddings with only one shared PLM encoder. The key idea is to introduce "relation prior tokens", which serve as priors for learning text embeddings. Specifically, relation prior tokens are prepended to the original text tokens and fed into a text encoder. The text encoder is encouraged to learn the shared knowledge across different relations, while the relation prior tokens are propelled to capture relation-specific signals. We further explore how to apply METERN to downstream tasks under different scenarios: (1) direct inference where the source relation is clearly singular (*e.g.,* using the "same-author" relation alone for author identification), and (2) indirect inference where source relations might be hybrid and need to be selected via learning. We evaluate METERN on five large-scale networks from the academic domain and the e-commerce domain with nine downstream tasks, where METERN outperforms competitive baseline methods significantly and consistently.

To summarize, our main contributions are as follows:

- Conceptually, we identify the semantic shift across different relations and formulate the problem of multiplex representation learning on text-rich networks.
- Methodologically, we propose METERN, which learns multiplex text representations with one text encoder and multiple relation prior tokens. Then, we introduce direct inference and "learn-to-select-source-relation" inference with METERN on different downstream tasks.
- Empirically, we conduct experiments on nine downstream tasks on five datasets from different domains, where METERN outperforms competitive baselines significantly and consistently.

## 2 PRELIMINARIES

### 2.1 MULTIPLEX TEXT-RICH NETWORKS

In a multiplex text-rich network, each node is associated with texts, and nodes are connected by multiple types of edges. We view the texts in each node as a document, and all such documents constitute a corpus $\mathcal{D}$.

**Definition 1** *(Multiplex Text-rich Networks) A multiplex text-rich network is defined as* $\mathcal{G} = \{\mathcal{G}^1, \mathcal{G}^2, ..., \mathcal{G}^{|\mathcal{R}|}\} = (\mathcal{V}, \mathcal{E}, \mathcal{D}, \mathcal{R})$*, where* $\mathcal{G}^r = (\mathcal{V}, \mathcal{E}^r, \mathcal{D})$ *is a network of the relation type* $r \in \mathcal{R}$*,* $\mathcal{V}$ *is the set of nodes,* $\mathcal{E} = \bigcup_{r \in \mathcal{R}} \mathcal{E}^r \subseteq \mathcal{V} \times \mathcal{V}$ *is the set of all edges,* $\mathcal{D}$ *is the set of documents, and* $\mathcal{R}$ *is the relation type set. Each node* $v_i \in \mathcal{V}$ *is associated with a document* $d_i \in \mathcal{D}$*. Note that* $|\mathcal{R}| > 1$ *for multiplex networks.*

### 2.2 PROBLEM DEFINITIONS

**Definition 2** *(Learning Multiplex Embeddings on Text-rich Networks) Given a multiplex text-rich network* $\mathcal{G} = (\mathcal{V}, \mathcal{E}, \mathcal{D}, \mathcal{R})$*, the task of learning multiplex embeddings is to build a model* $f_\Theta : \mathcal{V} \to \mathbb{R}^{|\mathcal{R}| \times d}$ *with parameters* $\Theta$ *to learn node representation vectors* $\boldsymbol{h}_{v_i} \in \mathbb{R}^{|\mathcal{R}| \times d}$ *for each node* $v_i \in \mathcal{V}$*, which should be able to be broadly utilized to various downstream tasks. Note that we aim to learn* $|\mathcal{R}|$ *embeddings for each node* $v_i$*, with each corresponding to a relation* $r \in \mathcal{R}$*.*

## 3 WHY MULTIPLEX EMBEDDINGS?

Learning a single embedding for each node/document presumes that the embedding $\boldsymbol{h}_v$ is enough to capture the semantics proximity between nodes with a similarity function $\mathrm{Sim}(\cdot)$, *i.e.*, $P(e_{ij}|v_i, v_j) \propto \mathrm{Sim}(\boldsymbol{h}_{v_i}, \boldsymbol{h}_{v_j})$, having an underlying assumption that all relations have analogous semantic distributions, *i.e.*, $P_{r_k}(e_{ij}|v_i, v_j) \approx P(e_{ij}|v_i, v_j) \approx P_{r_l}(e_{ij}|v_i, v_j)$ for different relations $r_k$ and $r_l$. However, this assumption does not always hold in real-world multiplex text-rich networks. For example, two papers $(v_i, v_j)$ written by the same author (relation: $r_k$) can be either published in the same venue (relation:$r_l$) or not; two items $(v_i, v_j)$ frequently purchased by the same user (relation: $r_k$) can have either the same brand (relation:$r_l$) or not.

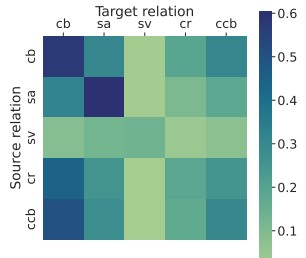

Figure 2: Distribution shift across different relations on a network of Geology papers. cb, sa, sv, cr, and ccb represent cited-by, same-author, same-venue, co-reference, and co-cited-by relation, respectively. Each entry is the PREC@1 of BERT embedding fine-tuned on the corresponding source relation distribution and tested on the corresponding target relation distribution.

In addition, we empirically find that the distribution shift across different relations, *i.e.*, $P_{r_k}(e_{ij}|v_i, v_j) \neq P_{r_l}(e_{ij}|v_i, v_j)$, in multiplex text-rich networks truly affects the learned embedding. We finetune BERT[1] (Devlin et al., 2019) to generate embeddings on one source relation distribution $P_{r_k}(e_{ij}|v_i, v_j)$ (row) and test the embeddings on the same or another target relation distribution $P_{r_l}(e_{ij}|v_i, v_j)$ (column). The results (measured by in-batch PREC@1) are shown in Figure 2. If the assumption of analogous distributions (*i.e.*, $P_{r_k}(e_{ij}|v_i, v_j) \approx P_{r_l}(e_{ij}|v_i, v_j)$) holds, the values in each cell should be nearly the same, which is not the case in Figure 2. As a result, if we only represent each node/text with one embedding vector, the single representation will mix up the semantics across different relation distributions and will lose accuracy in each of them. This inspires us to learn multiplex embeddings for each node/text, one for each relation, to capture the distinct semantics for each relation. The embeddings belonging to different relations can benefit different downstream tasks, *e.g.*, "same-author" for author identification, and "same-venue" for venue recommendation. More studies on the raw data distributions and learned embeddings can be found in Appendix A.4.

## 4 PROPOSED METHOD

A straightforward way to learn multiplex embeddings is to have multiple encoders (Jing et al., 2021; Park et al., 2020), one for each relation. However, this will make the total number of parameters $|\mathcal{R}|$ times as many as that of a single-encoder model, increasing both memory complexity and time complexity for training and inference, especially when every single encoder is computationally expensive (*e.g.*, pretrained language models (Devlin et al., 2019; Liu et al., 2019b)). From another perspective, the semantics of different relations can complement each other, *e.g.*, papers citing each other (*i.e.*, connected via the "cited-by" relation) are highly likely to be of similar topics, thus appearing in the same venue (*i.e.*, also connected via the "same-venue" relation). Nevertheless, adopting separate encoders for each relation will prevent the relations from benefiting each other. To this end, we propose to use only one text encoder to learn multiplex embeddings, with a simple but effective design of "relation prior tokens". We also propose flexible ways to apply the learned embeddings to downstream tasks, including direct inference for tasks where the source relation is clearly singular and "learn to select source relations" for tasks where the source relations might be hybrid. The whole model framework can be found in Figure 1.

### 4.1 METERN: LEARNING MULTIPLEX EMBEDDINGS WITH ONE TEXT ENCODER

**Representation Learning with Text Encoder.** Mainstream representation learning models (Cohan et al., 2020; Reimers & Gurevych, 2019) usually adopt a text encoder $\mathrm{Enc}(\cdot)$ (*e.g.*, Transformer (Vaswani et al., 2017)) to generate one node/text embedding for node $v_i$, *i.e.*, $\boldsymbol{h}_{v_i} = \mathrm{Enc}(d_i)$. Then, the predicted similarity score between node $v_i$ and $v_j$ is calculated by

$$P(e_{ij}|v_i, v_j) \propto \mathrm{Sim}(\boldsymbol{h}_{v_i}, \boldsymbol{h}_{v_j}) = \mathrm{Sim}(\mathrm{Enc}(d_i), \mathrm{Enc}(d_j)), \qquad (1)$$

where $\mathrm{Sim}(\cdot, \cdot)$ is a similarity measurement function, *e.g.*, dot product (Karpukhin et al., 2020) or cosine similarity (Reimers & Gurevych, 2019).

However, as discussed in Section 3, $P_{r_k}(e_{ij}|v_i, v_j) \neq P_{r_l}(e_{ij}|v_i, v_j)$ for different relations $r_k$ and $r_l$. This motivates us to obtain relation-conditioned embeddings $\boldsymbol{h}_{v|r}$ rather than just $\boldsymbol{h}_v$ to capture the diverse distribution $P_r(e)$ conditioned on relation $r$.

---

[1]We use the bert-base-uncased checkpoint.

**Generating Relation-conditioned Embeddings.** We represent the relations by introducing relation prior tokens $P_r = \{p_r^{(1)}, ..., p_r^{(m)}\}$ with embeddings $Z_r = \{\boldsymbol{z}_r^{(1)}, ..., \boldsymbol{z}_r^{(m)}\}$ for each relation $r$. Each $\boldsymbol{z}_r^{(t)} \in Z_r$ is a $k$-dimensional embedding vector for the relation prior token $p_r^{(t)} \in P_r$. To generate relation-conditioned embeddings with one encoder, we provide the encoder with both the node information and the target relation, *i.e.,* $d_i$ and $P_r$, as input. The representation $\boldsymbol{h}_{v_i|r}$ is obtained by

$$\boldsymbol{h}_{v_i|r} = \text{Enc}(\{P_r; d_i\}), \tag{2}$$

where $\{\cdot; \cdot\}$ is the token concatenation operation (Lester et al., 2021). It is worth noting that $P_r$ is specific for relation $r$ while the encoder $\text{Enc}(\cdot)$ is shared among different relations. As a result, the parameters in $Z_r$ will capture the relation-specific signals while the parameters in $\text{Enc}(\cdot)$ will learn the shared knowledge among different relations. Both the parameters in $\{Z_r\}_{r \in \mathcal{R}}$ and $\text{Enc}(\cdot)$ are learnable.

Then the relation prediction score is calculated by

$$P_r(e_{ij}|v_i, v_j) \propto \text{Sim}(\boldsymbol{h}_{v_i|r}, \boldsymbol{h}_{v_j|r}) = \text{Sim}(\text{Enc}(\{P_r; d_i\}), \text{Enc}(\{P_r; d_j\})). \tag{3}$$

In our experiment, we adopt the dot product as the similarity calculation function, *i.e.,* $\text{Sim}(\boldsymbol{h}_{v_i|r}, \boldsymbol{h}_{v_j|r}) = \boldsymbol{h}_{v_i|r} \cdot \boldsymbol{h}_{v_j|r}$.

**Multi-Relation Learning Objective.** During the unsupervised representation learning phase, all the relations will be learned simultaneously with the following log-likelihood:

$$\max_{\Theta} \mathcal{O} = \sum_{r \in \mathcal{R}} \sum_{e_{ij} \in \mathcal{E}^r} \log P_r(e_{ij}|v_i, v_j; \Theta), \tag{4}$$

Here, the conditional probability $P_r(e_{ij}|v_i, v_j; \Theta)$ is calculated as follows (Oord et al., 2018):

$$P_r(e_{ij}|v_i, v_j; \Theta) = \frac{\exp(\boldsymbol{h}_{v_i|r} \cdot \boldsymbol{h}_{v_j|r})}{\sum_{v_u \in \mathcal{V}} \exp(\boldsymbol{h}_{v_i|r} \cdot \boldsymbol{h}_{v_u|r})}, \tag{5}$$

To make the calculation efficient, we leverage the negative sampling technique (Jin et al., 2020; Mikolov et al., 2013) to simplify the objective and obtain our loss below.

$$\min_{\Theta} \mathcal{L} = \sum_{r \in \mathcal{R}} w_r \sum_{e_{ij} \in \mathcal{E}^r} -\log \frac{\exp(\boldsymbol{h}_{v_i|r} \cdot \boldsymbol{h}_{v_j|r})}{\exp(\boldsymbol{h}_{v_i|r} \cdot \boldsymbol{h}_{v_j|r}) + \sum_{v_u'} \exp(\boldsymbol{h}_{v_i|r} \cdot \boldsymbol{h}_{v_u'|r})} = \sum_{r \in \mathcal{R}} w_r \mathcal{L}_r. \tag{6}$$

In the equation above, $v_u'$ stands for a random negative sample. In our implementation, we use "in-batch negative samples" (Karpukhin et al., 2020) to reduce the encoding cost. Note that we add relation weights $w_r$ to control the relative learning speed of different relations (analysis can be found in Section 5.8).

### 4.2 INFERENCE WITH METERN

Node embeddings of high quality should be able to generalize to various downstream tasks within the given network scenario (Hamilton et al., 2017b). In this section, we propose to conduct direct inference and "learn-to-select-source-relations" inference with the learned multiplex embeddings.

**Direct Inference with an Evident Source Relation.** We propose direct inference for those downstream tasks where the semantically closest source relation $r_{\text{target}} \in \mathcal{R}$ to the target downstream task is clearly singular (*e.g.,* "same-author" relation for author identification and "same-venue" relation for venue recommendation). In these cases, we can directly infer the embedding of $r_{\text{target}}$ (with $P_{r_{\text{target}}}$) for the downstream task, without any downstream task training samples:

$$\boldsymbol{h}_{\text{target}} = \boldsymbol{h}_{v|r_{\text{target}}} = \text{Enc}(\{P_{r_{\text{target}}}; d\}). \tag{7}$$

**Learning to Select Source Relations.** We propose to learn to select source relations with some downstream task training samples when the semantically closest source relation is not clear (*e.g.,* paper classification and year prediction in academic networks; item classification and price prediction in e-commerce networks). We introduce a set of learnable query embeddings for the target task $Q_{\text{target}} = \{\boldsymbol{q}_{\text{target}}^{(1)}, \boldsymbol{q}_{\text{target}}^{(2)}, ..., \boldsymbol{q}_{\text{target}}^{(s)}\}$ to learn to select the source relation embeddings from $Z_{\mathcal{R}} = \{Z_r\}_{r \in \mathcal{R}}$ via attention-based mix-up, as follows:

$$\boldsymbol{z}_{\text{target}}^{(t)} = \sum_{\boldsymbol{z}_i \in Z_{\mathcal{R}}} \alpha_i^{(t)} \cdot \boldsymbol{z}_i, \qquad \alpha_i^{(t)} = \text{softmax}_{\boldsymbol{z}_i \in Z_{\mathcal{R}}}(\boldsymbol{z}_i \cdot \boldsymbol{q}_{\text{target}}^{(t)}), \tag{8}$$

$$Z_{\text{target}} = \{\boldsymbol{z}_{\text{target}}^{(1)}, \boldsymbol{z}_{\text{target}}^{(2)}, ..., \boldsymbol{z}_{\text{target}}^{(s)}\}, \qquad P_{\text{target}} = \{p_{\text{target}}^{(1)}, p_{\text{target}}^{(2)}, ..., p_{\text{target}}^{(s)}\}, \tag{9}$$

$$\boldsymbol{h}_{\text{target}} = \text{Enc}(\{P_{\text{target}}; d\}). \tag{10}$$

$Z_{\mathcal{R}} = \{Z_r\}_{r \in \mathcal{R}}$ is the source relation embedding pool (with $|\mathcal{R}| \times m$ embeddings), which is the union of all the source relation embeddings for $r \in \mathcal{R}$. We use $P_{\text{target}}$ and $Z_{\text{target}}$ to represent the downstream task prior tokens and their embeddings, respectively (i.e., $\boldsymbol{z}_{\text{target}}^{(t)}$ is the embedding for $p_{\text{target}}^{(t)}$). Note that only the parameters in $Q_{\text{target}}$ are trainable to learn to select source relations.

With the mechanism above, the model will learn to assign a larger weight $\alpha$ to the relation embedding $\boldsymbol{z}_i$ which can contribute more to solving the downstream task, thus learning to select source relations for downstream tasks. At the same time, the learned weight $\alpha$ can also help reveal the hidden correlation between relations and tasks (see the analysis in Section 5.5 and Appendix A.7).

### 4.3 DISCUSSIONS

**Complexity Analysis.** In our experiment, we use the Transformer encoder (Vaswani et al., 2017) as the Enc(·) function. *Time Complexity:* Given a node $v_i$ associated with a document $d_i$ containing $p$ tokens, the time complexity of our multiplex representation learning model with $m$ embeddings for each relation is $\mathcal{O}((p + m)^2)$, which is on par with the $\mathcal{O}(p^2)$ complexity of the single relation representation learning model since $m \ll p$. *Memory Complexity:* Given a network with $|\mathcal{R}|$ types of relations and $T$ parameters in the Enc(·), the parameter complexity of our multiplex representation learning model is $\mathcal{O}(T + |\mathcal{R}|mk)$, which is nearly the same as the $\mathcal{O}(T)$ complexity of the single relation representation learning model since $|\mathcal{R}|mk \ll T$. The empirical time efficiency study and memory efficiency study are shown in Section 5.6.

**Difference from Existing Works (Lester et al., 2021; Liu et al., 2021; Qin & Eisner, 2021).** These works propose to add learnable "prompt tokens" to the text sequences before feeding them into the Transformer architecture, sharing a similar design with our relation prior tokens. However, there are three inherent differences between these works and our work: (1) Different focuses: Existing works (Lester et al., 2021; Liu et al., 2021; Qin & Eisner, 2021) focus more on efficient language model tuning (*i.e.*, training the prompt embedding only), while we focus on joint representation learning (*i.e.*, using the relation embeddings to capture relation-specific information and the shared encoder to capture knowledge shared across relations). (2) Different designs: The prompt tokens used in NLP problems (Lester et al., 2021; Liu et al., 2021; Qin & Eisner, 2021) generally correspond to natural language descriptions of the tasks and labels. In our case, we use relation prior tokens to encode the abstract relations between documents that may not be directly described by natural language tokens. As a result, the representation of an unseen relation may need to be learned as a mixture of source relation representations instead of simply being represented by natural language prompts. (3) Different applications: Existing works focus on NLU (Liu et al., 2021; Qin & Eisner, 2021), NLG (Lester et al., 2021), and knowledge probing (Qin & Eisner, 2021) tasks, while our work focuses on multiplex representation learning.

## 5 EXPERIMENTS

In this section, we first introduce the five datasets. Then, we demonstrate the high quality of the learned multiplex embeddings by METERN. After that, we show the effectiveness of METERN on downstream tasks with direct source relation inference and "learn-to-select-source-relation" inference, respectively. Finally, we conduct efficiency analysis, multiplex embedding visualization, and the study of the relation weight $w_r$. More experiments on relation token embedding initialization and relation token embedding visualization can be found in Appendices A.6 and A.8, respectively.

### 5.1 DATASETS

We run experiments on both academic networks from the Microsoft Academic Graph (MAG) (Sinha et al., 2015; Zhang et al., 2023) and e-commerce networks from Amazon (He & McAuley, 2016). In academic networks, nodes correspond to papers and there are five types of relations among papers: "cited-by" (cb), "same-author" (sa), "same-venue" (sv), "co-reference" (cr), and "co-cited-by" (ccb); while in e-commerce networks, nodes are items and there are four types of relations between items: "co-purchased" (cop), "co-viewed" (cov), "bought-together" (bt), and "co-brand" (cob). Since both MAG and Amazon have multiple domains, we select two domains from MAG and three domains from Amazon. In total, we have five datasets in the experiments (*i.e.,* MAG-Geology, MAG-Mathematics, Amazon-Clothes, Amazon-Home, and Amazon-Sports). The datasets' statistics can be found in Table 7.

### 5.2 BASELINES

We compare METERN with three kinds of baselines, large-scale corpora finetuned text embedders, multiplex graph neural networks, and multi-relation learning language models. The first category includes SPECTER (Cohan et al., 2020), SciNCL (Ostendorff et al., 2022), Sentence-Transformer

Table 1: Multiplex representation learning experiments on academic networks: Geology and Mathematics. cb, sa, sv, cr, and ccb represent "`cited-by`", "`same-author`", "`same-venue`", "`co-reference`", and "`co-cited-by`" relation, respectively.

| Model | Geology | | | | | | Mathematics | | | | | |
|---|---|---|---|---|---|---|---|---|---|---|---|---|
| | cb | sa | sv | cr | ccb | Avg. | cb | sa | sv | cr | ccb | Avg. |
| SPECTER | 12.84 | 12.89 | 1.5 | 5.56 | 9.1 | 8.38 | 28.74 | 23.55 | 2.39 | 15.96 | 25.59 | 19.25 |
| SciNCL | 15.91 | 14.3 | 1.57 | 6.41 | 10.4 | 9.72 | 36.14 | 26.41 | 2.83 | 19.82 | 30.69 | 23.18 |
| MPNet-v2 | 30.87 | 20.94 | 1.94 | 10.36 | 17.16 | 16.25 | 46.12 | 29.92 | 3.11 | 23.60 | 36.42 | 27.83 |
| OpenAI-ada-002 | 30.39 | 21.08 | 2.02 | 16.57 | 16.69 | 17.35 | 39.86 | 27.22 | 2.67 | 19.81 | 31.62 | 24.24 |
| DMGI | 28.99 | 27.79 | 4.91 | 9.86 | 16.32 | 17.58 | 46.55 | 42.62 | 6.11 | 27.80 | 38.87 | 28.85 |
| HDMI | 37.89 | 34.87 | 3.63 | 11.32 | 19.55 | 21.45 | 52.65 | 52.71 | 5.54 | 31.80 | 42.54 | 37.05 |
| Vanilla FT | 54.42 | 43.20 | 5.95 | 18.48 | 29.93 | 30.40 | 75.03 | 63.46 | 8.71 | 44.76 | 59.94 | 50.38 |
| MTDNN | 58.40 | 52.50 | 10.60 | 19.81 | 31.61 | 34.58 | 78.18 | 71.04 | 12.90 | 47.39 | 61.75 | 54.25 |
| Ours | **60.33** | **55.55** | **12.30** | **20.71** | **32.92** | **36.36** | **79.40** | **72.51** | **14.03** | **47.81** | **62.24** | **55.20** |

Table 2: Multiplex representation learning experiments on e-commerce networks: Clothes, Home, and Sports. cop, cov, bt, and cob represent "`co-purchased`", "`co-viewed`", "`bought-together`", and "`co-brand`" relation, respectively.

| Model | Clothes | | | | | Home | | | | | Sports | | | | |
|---|---|---|---|---|---|---|---|---|---|---|---|---|---|---|---|
| | cop | cov | bt | cob | Avg. | cop | cov | bt | cob | Avg. | cop | cov | bt | cob | Avg. |
| MPNet-v2 | 55.89 | 60.92 | 59.75 | 39.12 | 53.92 | 52.02 | 61.83 | 62.04 | 38.10 | 53.50 | 41.60 | 64.61 | 49.82 | 40.61 | 49.16 |
| OpenAI-ada-002 | 65.30 | 70.87 | 69.44 | 48.32 | 63.48 | 60.99 | 71.43 | 71.36 | 47.86 | 62.91 | 50.80 | 73.70 | 60.20 | 54.06 | 59.69 |
| DMGI | 56.10 | 52.96 | 58.46 | 30.88 | 49.60 | 48.27 | 52.74 | 57.90 | 48.81 | 51.93 | 41.37 | 46.27 | 41.24 | 31.92 | 40.20 |
| HDMI | 62.85 | 63.00 | 69.69 | 52.50 | 62.01 | 51.75 | 57.91 | 57.91 | 53.39 | 55.24 | 45.43 | 61.22 | 55.56 | 52.66 | 53.72 |
| Vanilla FT | 81.57 | 80.46 | 88.52 | 67.38 | 79.48 | **73.72** | 75.49 | 85.80 | 76.83 | 77.96 | **68.22** | 77.11 | 80.78 | 78.46 | 76.14 |
| MTDNN | 80.30 | 78.75 | 87.58 | 65.94 | 78.14 | 72.49 | 75.17 | 84.00 | 77.29 | 77.24 | 66.20 | 76.50 | 79.72 | 78.69 | 75.28 |
| Ours | **82.04** | **81.18** | **88.90** | **68.34** | **80.12** | 73.59 | **79.06** | **86.58** | **80.07** | **79.83** | 67.92 | **79.85** | **81.52** | **81.54** | **77.71** |

(Reimers & Gurevych, 2019), and the OpenAI-ada-002 embedder (Brown et al., 2020). SPECTER (Cohan et al., 2020) is a text embedder finetuned from SciBERT (Beltagy et al., 2019) with citation-guided contrastive learning. SciNCL (Ostendorff et al., 2022) further improves SPECTER by introducing controlled nearest neighbor sampling. For Sentence-Transformer (Reimers & Gurevych, 2019), we use the recent best checkpoint[2] which finetunes MPNet (Song et al., 2020) with over 1 billion sentence pairs from 32 domains. OpenAI-ada-002[3] is the recent large language model-based text embedder proposed by OpenAI. The second category includes DMGI (Park et al., 2020) and HDMI (Jing et al., 2021). DMGI (Park et al., 2020) is a graph convolutional network-based multiplex node embedding encoder. HDMI (Jing et al., 2021) further improves DMGI by proposing an alternative semantic attention-based fusion module. For multiplex GNN methods, we use bag-of-words embeddings as the initial node feature vectors. The third category includes the vanilla finetuned language model (Vanilla FT) and MTDNN (Liu et al., 2019a). For Vanilla FT, we finetuned BERT (Devlin et al., 2019) with all the relation pairs, without distinguishing among them. In this way, the model will only output one single embedding for each node. MTDNN (Liu et al., 2019a) is a multi-task learning model with a shared language model backbone for different tasks and task-specific modules connected to the backbone. As a result, it can output multiplex representations for each node corresponding to different tasks (relations). Note that Vanilla FT, MTDNN, and METERN are all initialized by the same bert-base-uncased checkpoint.

## 5.3 MULTIPLEX REPRESENTATION LEARNING

We test the quality of the generated embeddings of different models on multiplex relation prediction tasks on the network. Given a query node/text (*e.g.,* a paper or an item), a target relation (*e.g.,* cb, sa, cop, or cov), and a candidate node/text list (*e.g.,* papers or items), we aim to predict which key node in the candidate list should be linked to the given query node under the target relation. We use PREC@1 as the metric. More detailed information on experimental settings can be found in Appendix A.5.1

The results on academic networks and e-commerce networks are shown in Table 1 and Table 2, respectively. From the result, we can find that: 1) METERN performs significantly and consistently better than all the baseline methods on all types of relation prediction on all datasets (except cop prediction on Home and Sports). 2) In academic networks, multiplex representation learning methods (MTDNN and METERN) can always outperform the single representation learning method (Vanilla FT); while in e-commerce networks, Vanilla FT performs on par with MTDNN and METERN. This is because the relations in e-commerce networks are semantically closer to each other, while the

---

[2]https://huggingface.co/sentence-transformers/all-mpnet-base-v2
[3]https://openai.com/blog/new-and-improved-embedding-model

Table 3: Direct inference with an evident source relation (no task-specific training) on academic networks and e-commerce networks.

| Model | Geology | | Mathematics | | Clothes | Home | Sports |
| --- | --- | --- | --- | --- | --- | --- | --- |
| | Venue-Rec | Author-Idf | Venue-Rec | Author-Idf | Brand-Pred | Brand-Pred | Brand-Pred |
| SPECTER | 4.06 | 18.28 | 4.54 | 23.48 | - | - | - |
| SciNCL | 5.14 | 24.58 | 6.51 | 35.49 | - | - | - |
| MPNet-v2 | 6.96 | 46.99 | 7.1 | 47.51 | 43.75 | 57.41 | 61.22 |
| OpenAI-ada-002 | 6.20 | 45.35 | 6.07 | 42.74 | 59.09 | 63.59 | 69.14 |
| DMGI | 9.21 | 40.91 | 9.57 | 49.37 | 49.60 | 41.71 | 40.63 |
| HDMI | 2.96 | 20.36 | 3.44 | 25.55 | 53.07 | 58.30 | 51.07 |
| Vanilla FT | 11.72 | 62.76 | 13.35 | 68.31 | 66.16 | 74.36 | 76.87 |
| MTDNN | 13.63 | 63.72 | **15.03** | 68.45 | 63.79 | 77.98 | 78.86 |
| Ours | **14.44** | **68.43** | 14.64 | **71.63** | **66.69** | **79.50** | **81.87** |

Table 4: Learning to select source relations on academic networks. ($\uparrow$) means the greater the score is, the better the model is, and ($\downarrow$) otherwise.

| Model | Geology | | | | Mathematics | | | |
| --- | --- | --- | --- | --- | --- | --- | --- | --- |
| | Paper-Rec ($\uparrow$) | Paper-Cla ($\uparrow$) | Citation-Pred ($\downarrow$) | Year-Pred ($\downarrow$) | Paper-Rec ($\uparrow$) | Paper-Cla ($\uparrow$) | Citation-Pred ($\downarrow$) | Year-Pred ($\downarrow$) |
| Vanilla FT | $76.45_{0.00}$ | $41.62_{0.04}$ | $16.28_{0.01}$ | $9.01_{0.01}$ | $84.94_{0.00}$ | $36.76_{0.18}$ | $9.16_{0.01}$ | $9.77_{0.00}$ |
| MTDNN | $77.99_{0.02}$ | $45.84_{0.48}$ | $15.76_{0.04}$ | $8.63_{0.12}$ | $85.42_{0.06}$ | $38.47_{0.19}$ | $9.13_{0.28}$ | $9.74_{0.00}$ |
| Ours | $80.58_{0.00}$ | $46.68_{0.08}$ | $16.00_{0.09}$ | $8.34_{0.01}$ | $86.56_{0.03}$ | $39.20_{0.29}$ | $8.95_{0.02}$ | $9.26_{0.03}$ |

relations in academic networks are more dissimilar or even conflict with each other (see Figure 2 and Figure 7). 3) Although SPECTER (Cohan et al., 2020), SciNCL (Ostendorff et al., 2022), Sentence-Transformer (Reimers & Gurevych, 2019), and OpenAI-ada-002 (Brown et al., 2020) are finetuned on large-scale corpora, they perform badly compared with METERN since they only output one embedding which cannot capture the diverse relation semantics. 4) Multiplex GNN methods (DMGI and HDMI) can utilize neighbor information from the network to output multiplex embeddings but fail to capture the diverse and contextualized text semantics associated with nodes.

## 5.4 DIRECT INFERENCE WITH AN EVIDENT SOURCE RELATION

We conduct experiments on venue recommendation (Venue-Rec) and author identification (Author-Idf) in academic networks and brand prediction (Brand-Pred) in e-commerce networks to verify the effectiveness of METERN's direct inference ability with an evident source relation. All three tasks are formulated as matching tasks where we use PREC@1 as the metric. For Venue-Rec, Author-Idf, and Brand-Pred, we use the sv, sa, and cob representation, respectively.

The results are shown in Table 3. From the results, we can find that: METERN outperforms all the baseline methods significantly and consistently on all the tasks (except Venue-Rec on Mathematics), which demonstrates that the multiplex embeddings generated by METERN are of good quality and can be directly applied to downstream tasks. Detailed information about the tasks and experimental settings can be found in Appendix A.5.2.

## 5.5 LEARNING TO SELECT SOURCE RELATIONS

We perform experiments on paper recommendation, paper classification, citation prediction, and year prediction on academic networks, and item classification and price prediction on e-commerce networks to test if METERN can learn to select source relations. We use PREC@1 as the metric for paper recommendation, Macro-F1 as the metric for paper classification and item classification, and RMSE as the metric for citation prediction, year prediction, and price prediction.

The results on academic networks and e-commerce networks are shown in Table 4 and Table 5, respectively. We also show the learned source relation weight for citation prediction and paper recommendation in Figure 3. From the results, we can find that: 1) METERN outperforms all the baseline methods



(a) Citation    (b) Rec

Figure 3: The learned source relation weights for citation prediction and paper recommendation on Geology network.

significantly and consistently on all the tasks (except citation prediction on Geology and item classification on Sports). 2) METERN can learn to select different relations for different downstream tasks, *i.e.,* "same-author" and "same-venue" for citation prediction (authors and venues can

Table 5: Learning to select source relations on e-commerce networks.

| Model | Clothes | | Home | | Sports | |
|---|---|---|---|---|---|---|
| | Item-Cla ($\uparrow$) | Price-Pred ($\downarrow$) | Item-Cla ($\uparrow$) | Price-Pred ($\downarrow$) | Item-Cla ($\uparrow$) | Price-Pred ($\downarrow$) |
| Vanilla FT | $87.84_{0.08}$ | $17.88_{0.02}$ | $90.68_{0.15}$ | $76.41_{0.03}$ | $\mathbf{80.94}_{0.34}$ | $20.27_{0.00}$ |
| MTDNN | $86.34_{0.38}$ | $18.53_{0.16}$ | $90.62_{0.20}$ | $75.07_{0.06}$ | $78.76_{0.14}$ | $21.28_{0.05}$ |
| Ours | $\mathbf{88.19}_{0.22}$ | $\mathbf{17.73}_{0.01}$ | $\mathbf{92.25}_{0.15}$ | $\mathbf{74.55}_{0.05}$ | $80.37_{0.22}$ | $\mathbf{20.01}_{0.02}$ |

Table 6: Time and memory costs on Geology.

| Attribute | Vanilla FT | MTDNN | METERN |
|---|---|---|---|
| Time | 15h 35min | 16h 23min | 17h 20min |
| Memory | 24,433 MB | 26,201MB | 28,357 MB |

largely determine the citation of the paper), and "`cited-by`" and "`co-cited-by`" for paper recommendation (papers connected via "`cited-by`" or "`co-cited-by`" relations are more likely to be informative to each other's readers). Detailed information about the tasks and experimental settings can be found in Appendix A.5.3. More result analysis can be found in Appendix A.7.

## 5.6 EFFICIENCY ANALYSIS

We run experiments on Geology to study the time complexity and memory complexity of METERN, comparing with Vanilla FT and MTDNN. All compared models are trained for 40 epochs on four NVIDIA A6000 GPU devices with a total training batch size set as 512. We show the result in Table 6. From the result, we can find that the time complexity and memory complexity of training METERN are on par with those of training Vanilla FT and MTDNN.

## 5.7 MULTIPLEX EMBEDDING VISUALIZATION

We visualize the multiplex node embeddings $h_{v|r}$ learned by METERN with t-SNE (Van der Maaten & Hinton, 2008). The results on Geology are shown in Figure 4. We randomly select one center node from the network, together with its neighboring nodes of the "`cited-by`" (cb), "`same-author`" (sa), and "`same-venue`" (sv) relations. Then, we utilize METERN to encode all the nodes (including the center node and all neighboring nodes) in different relation representation spaces (*i.e.,* to obtain $h_{v|r_{cb}}$, $h_{v|r_{sa}}$, and $h_{v|r_{sv}}$). In the figure, neighboring nodes of different relations are marked in different colors and shapes. From Figure 4, we can find that, in the output embedding space corresponding to one relation (*e.g.*, the $h_{v|r_{cb}}$ latent space corresponding to "`cited-by`" in Figure 4(a)), the center node (the blue circle)'s neighboring nodes of the corresponding relation (*e.g.*, "`cited-by`" relation neighbors, marked as orange rectangles) are close to the center node. This demonstrates that METERN can capture the semantics of different relations and represent them in different latent subspaces.

## 5.8 ANALYSIS OF RELATION WEIGHT $w_r$

In METERN, different relations sometimes have different learning speeds, and $w_r$ in Eq.(6) can be tuned to balance the learning speed of different relations. We analyze the effect of $w_r$ by showing the embedding performance on the validation set of two different weight settings (uniform and tuned) on the Geology network. The results are shown in Figure 5. In the uniform setting, all relations have the same weight (*i.e.*, $[w_{cb}, w_{sa}, w_{sv}, w_{cr}, w_{ccb}] = [1, 1, 1, 1, 1]$). We can find that as the learning step increases, cb, cr, and ccb embeddings quickly reach the best performance and then overfit, while sa and sv embeddings are still underfitting. This motivates us to increase the weight for sa and sv since their learning speeds are slow. We can find that when we use the tuned weight set ($[w_{cb}, w_{sa}, w_{sv}, w_{cr}, w_{ccb}] = [1, 2, 2, 1, 1]$), the learning speeds of different relations are more balanced, leading to a generally better multiplex encoder.

# 6 RELATED WORK

## 6.1 LEARNING EMBEDDINGS ON MULTIPLEX NETWORKS

Multiplex networks are also referred to as multi-view networks (Qu et al., 2017; Shi et al., 2018) and multi-dimensional networks (Berlingerio et al., 2013; Ma et al., 2018) in the literature. Multiplex networks consist of multiple relation/edge types among a set of single-typed nodes. They can be viewed as special cases of heterogeneous networks (Dong et al., 2020; Sun et al., 2011) where nodes only have one type. The first category of existing works propose to use one embedding vector to represent each node (Dong et al., 2017; Hu et al., 2020; Wang et al., 2019; Zhang et al., 2019). For example, HetGNN (Zhang et al., 2019) introduces a graph neural network that utilizes heterogeneous neighbor aggregation to capture the diverse relation semantics; HAN (Wang et al., 2019) proposes two-level attention including node-level and semantic-level to conduct relation semantics encoding; HGT (Hu et al., 2020) introduces a more complex heterogeneous graph Transformer architecture to

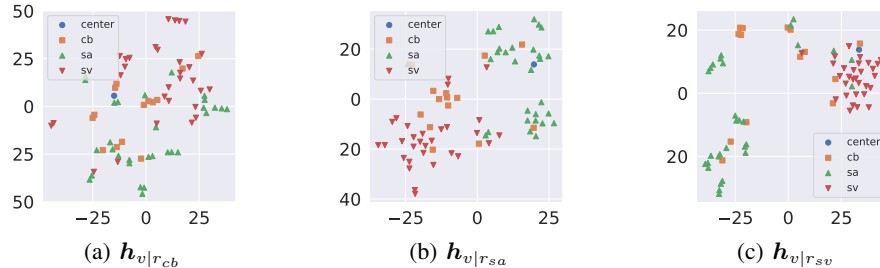

Figure 4: Multiplex embedding visualization on Geology network. cb, sa, and sv represent the "cited-by", "same-author", and "same-venue" relation respectively.

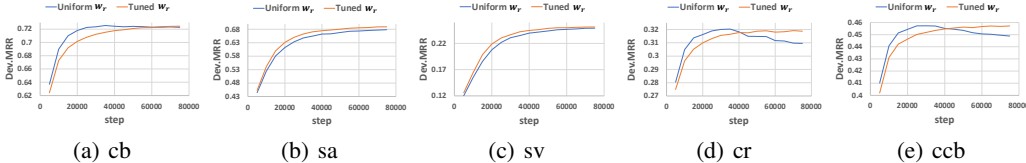

(a) cb    (b) sa    (c) sv    (d) cr    (e) ccb

Figure 5: Analysis of relation weights $w_r$ on Geology. The x-axis is the training step and the y-axis is MRR on the validation set. We compare uniform $w_r$ with a better-tuned $w_r$.

conduct node and edge semantics joint encoding. However, they all have an assumption that one embedding is enough to capture the semantics of all relations, which does not quite hold as shown in Section 3. The second category of works try to develop multiplex embeddings for each node (Jing et al., 2021; Park et al., 2020; Qu et al., 2017; Zhang et al., 2018), with one for each relation. For instance, MNE (Zhang et al., 2018) proposes a random walk-based multiplex embedding learning method; MVE (Qu et al., 2017) introduces a collaboration framework to capture the individual view network semantics and integrate them for the robust representations; DMGI (Park et al., 2020) proposes a deep infomax framework to conduct multiplex embedding learning with graph neural networks; HDMI (Jing et al., 2021) further enhances DMGI by introducing high-order mutual information with a fusion module. However, those works emphasize more on network structure encoding, while do not take the rich textual information associated with nodes into consideration.

### 6.2 TEXT EMBEDDINGS

Text embeddings (Brown et al., 2020; Cohan et al., 2020; Harris, 1954; Le & Mikolov, 2014; Ostendorff et al., 2022; Reimers & Gurevych, 2019) effectively capture the textual semantic similarity between text units (*i.e.,* sentences and documents) via distributed representation learning. Early work (Harris, 1954) proposes the bag-of-words vector space model, representing a text unit as a multiset of its words, disregarding grammar and even word order. Paragraph Vector (Le & Mikolov, 2014) is then introduced to capture the text semantics rather than just word appearance by representing each document with a dense vector trained to predict words in the document. As large-scale pretrained language models (Beltagy et al., 2019; Brown et al., 2020; Devlin et al., 2019; Liu et al., 2019b) are proposed, Sentence-BERT (Reimers & Gurevych, 2019) further finetunes BERT (Devlin et al., 2019) and RoBERTa (Liu et al., 2019b) by using a siamese and triplet network structure to derive semantically meaningful text embeddings which can measure text proximity with the cosine similarity between embeddings. OpenAI-ada-002 is recently developed as a powerful text embedding model based on the GPT series of large language models (Brown et al., 2020). Specific to the academic domain, SPECTER (Cohan et al., 2020) finetunes SciBERT by introducing positive and negative paper pairs, while SciNCL (Ostendorff et al., 2022) further enhances SPECTER by developing a nearest neighbor sampling strategy. However, all the existing works presume that one embedding can capture the general semantics for each text unit and do not take the diverse text relation semantics into consideration.

### 7 CONCLUSIONS

We discuss the limitations of our work in Appendix A.1. In this work, we tackle the problem of representation learning on multiplex text-rich networks. To this end, we introduce the METERN framework to learn multiplex text representations with only one text encoder. METERN introduces relation prior tokens to capture the relation-specific signals and one text encoder to model the shared knowledge across relations. We conduct experiments on nine downstream tasks and five networks from two domains, where METERN outperforms baselines significantly and consistently. Interesting future directions include: (1) exploring other more advanced graph-empowered text encoders for learning multiplex embeddings, and (2) applying the framework to more network-related tasks such as network generation conditioned on various relations.

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

# A   APPENDIX

## A.1   LIMITATIONS

In this work, we mainly focus on learning multiplex text/node representations on text-rich networks and solving downstream tasks (*e.g.,* classification, regression, matching) with the learned embeddings. Because of the limited computational budgets, our PLM text encoder (bert-base-uncased) is medium-scale. In the future, we will explore applying a similar multiplex representation learning philosophy to large-scale language models. Other interesting future directions include designing more advanced graph-empowered language models to learn the multiplex embeddings and adopting the model into more real-world applications such as network generation.

## A.2   ETHICAL CONSIDERATIONS

PLMs have been shown to be highly effective in encoding contextualized semantics and understanding documents, as evidenced by several studies (Devlin et al., 2019; Liu et al., 2019b; Clark et al., 2020). However, some researchers have pointed out certain limitations associated with these models, including the presence of social bias (Liang et al., 2021) and the propagation of misinformation (Abid et al., 2021). In our work, we focus on utilizing the relation signals between texts from the multiplex text-rich network structure to facilitate the understanding of the semantics of the texts, which we believe could help to address issues related to bias and misinformation.

## A.3   DATASETS

The statistics of the five datasets can be found in Table 7. In academic networks, nodes correspond to papers and there are five types of relations between papers: "`cited-by`" (cb), "`same-author`" (sa), "`same-venue`" (sv), "`co-reference`" (cr), and "`co-cited-by`" (ccb); while in e-commerce networks, nodes are items and there are four types of relations between items: "`co-purchased`" (cop), "`co-viewed`" (cov), "`bought-together`" (bt), and "`co-brand`" (cob).

Table 7: Dataset Statistics.

| Dataset | #Nodes | #Relations (Edges) |
|---------|--------|--------------------|
| Geology | 431,834 | cb (1,000,000), sa (1,000,000), sv (1,000,000) 
 cr (1,000,000), ccb (1,000,000) |
| Mathematics | 490,551 | cb (1,000,000), sa (1,000,000), sv (1,000,000) 
 cr (1,000,000), ccb (1,000,000) |
| Clothes | 208,318 | cop (100,000), cov (100,000) 
 bt (100,000), cob (50,000) |
| Home | 192,150 | cop (100,000), cov (100,000) 
 bt (50,000), cob (100,000) |
| Sports | 189,526 | cop (100,000), cov (100,000) 
 bt (50,000), cob (100,000) |

## A.4   DISTRIBUTION SHIFT BETWEEN DIFFERENT RELATIONS

In Section 3, we show the learned embedding distribution shift between different relations on Geology in Figure 2. In this section, we calculate the raw data distribution shift between different relations' distribution $P_{r_k}(e_{ij}|v_i, v_j)$. The distribution shift is measured by Jaccard score:

$$\text{Jac}(r_k, r_l) = \frac{|P_{r_k}(e_{ij}) \cap P_{r_l}(e_{ij})|}{|P_{r_k}(e_{ij}) \cup P_{r_l}(e_{ij})|} \tag{11}$$

Since the whole networks are too large to calculate the Jaccard score, we randomly sample a sub-network from each network that contains 10,000 nodes and calculate Eq.(A.4). The results on Geology, Mathematics, Clothes, Home and Sports networks can be found in Figure 6. If the

assumption of analogous distributions (*i.e.*, $P_{r_k}(e_{ij}|v_i, v_j) \approx P_{r_l}(e_{ij}|v_i, v_j)$) holds, the values in each cell should be nearly one, which is not the case in Figure 6.

More empirical experiments on the learned embedding distribution shift between relations in Mathematics, Clothes, Home, and Sports networks can be found in Figure 7. We finetune BERT[4] (Devlin et al., 2019) to generate embeddings on one source relation distribution $P_{r_k}(e_{ij}|v_i, v_j)$ (row) and test the embeddings on the same or another target relation distribution $P_{r_l}(e_{ij}|v_i, v_j)$ (column). If the assumption of analogous distributions (*i.e.*, $\tilde{P}_{r_k}(e_{ij}|v_i, v_j) \approx P_{r_l}(e_{ij}|v_i, v_j)$) holds, the values in each cell should be nearly the same, which is not the case in Figure 7.

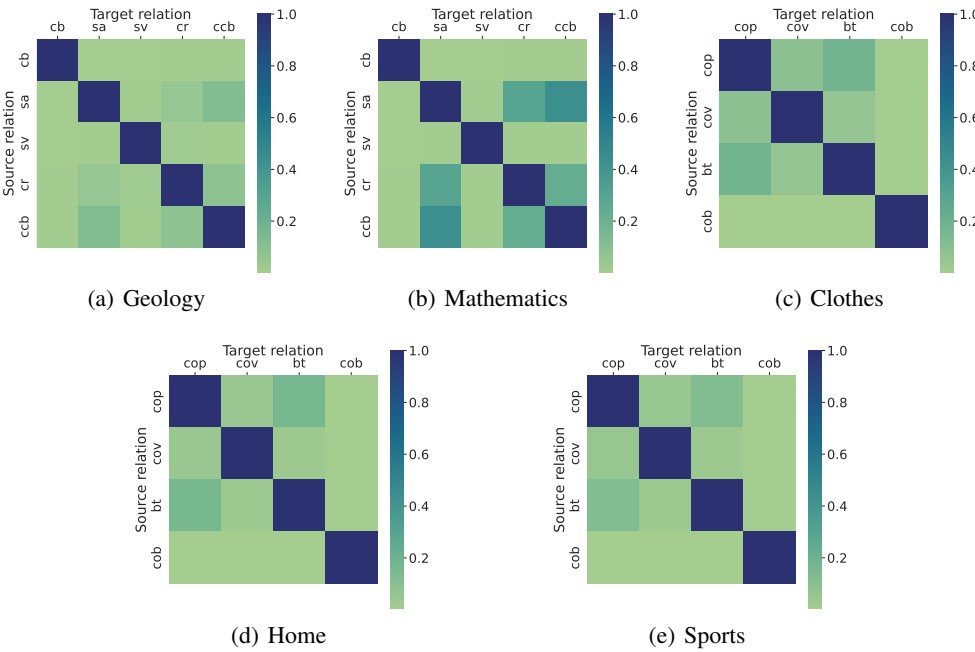

Figure 6: Raw data distribution shift between different relations on Geology, Mathematic, Clothes, Home, and Sports network. cb, sa, sv, cr, and ccb represent "`cited-by`", "`same-author`", "`same-venue`", "`co-reference`", and "`co-cited-by`" relation respectively. cop, cov, bt, and cob represent "`co-purchased`", "`co-viewed`", "`bought-together`", and "`co-brand`" relation respectively. Each entry is the Jaccard score between the corresponding two relation distributions.

## A.5 EXPERIMENTAL SETTING

### A.5.1 MULTIPLEX REPRESENTATION LEARNING

**Hyperparameter setting.** To facilitate the reproduction of our representation learning experiments, we provide the hyperparameter configuration in Table 8. Vanilla FT, MTDNN, and METERN use exactly the same set of hyperparameters for a fair comparison. The last layer `[CLS]` token hidden states are utilized to develop $\boldsymbol{h}_{v|r}$ for Vanilla FT, MTDNN, and METERN. Paper titles and item titles are used as text associated with the nodes in the two kinds of networks, respectively. (For some items, we concatenate the item title and description together since the title is too short.) The models are trained for 40 epochs on 4 Nvidia A6000 GPUs with a total batch size of 512. The total time cost is around 17 hours and 2 hours for networks in the academic domain and e-commerce domain respectively. Code is available at `https://anonymous.4open.science/r/METER-submit-2C7B`.

---

[4]We use the bert-base-uncased checkpoint.

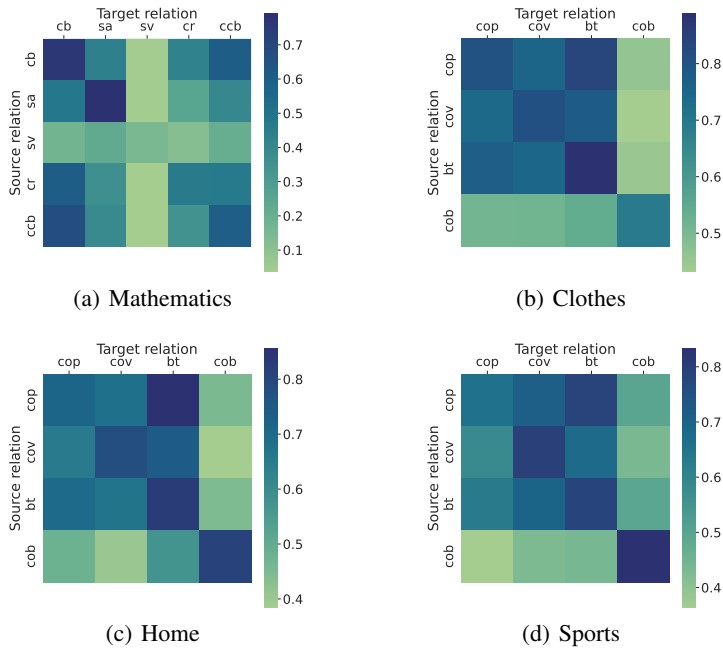

Figure 7: Distribution shift between different relations on Mathematic, Clothes, Home, and Sports network. cb, sa, sv, cr, and ccb represent "`cited-by`", "`same-author`", "`same-venue`", "`co-reference`", and "`co-cited-by`" relation respectively. cop, cov, bt, and cob represent "`co-purchased`", "`co-viewed`", "`bought-together`", and "`co-brand`" relation respectively. Each entry is the PREC@1 of BERT fine-tuned on the corresponding source relation distribution and tested on the corresponding target relation distribution.

Table 8: Hyper-parameter configuration for representation learning.

| Parameter | Geology | Mathematics | Clothes | Home | Sports |
|---|---|---|---|---|---|
| Max Epochs | 40 | 40 | 40 | 40 | 40 |
| Peak Learning Rate | 5e-5 | 5e-5 | 5e-5 | 5e-5 | 5e-5 |
| Batch Size | 512 | 512 | 512 | 512 | 512 |
| # Prior Tokens $m$ | 5 | 5 | 5 | 5 | 5 |
| Warm-Up Epochs | 4 | 4 | 4 | 4 | 4 |
| Sequence Length | 32 | 32 | 32 | 32 | 32 |
| Adam $\epsilon$ | 1e-8 | 1e-8 | 1e-8 | 1e-8 | 1e-8 |
| Adam $(\beta_1, \beta_2)$ | (0.9, 0.999) | (0.9, 0.999) | (0.9, 0.999) | (0.9, 0.999) | (0.9, 0.999) |
| Clip Norm | 1.0 | 1.0 | 1.0 | 1.0 | 1.0 |
| Dropout | 0.1 | 0.1 | 0.1 | 0.1 | 0.1 |

A.5.2 DIRECT INFERENCE WITH AN EVIDENT SOURCE RELATION

We provide problem definitions and experimental settings for tasks in Section 5.4. The tasks include venue recommendation, author identification, and brand prediction.

**Venue Recommendation.**

*Problem Definition.* Given a query paper node (with associated text) and a candidate venue list (each with its published papers), we aim to predict which venue in the candidate list should be recommended for the given query paper.

*Experimental Settings.* We adopt in-batch testing with a testing batch size of 256. We use PREC@1 as the metric. The max sequence length is 32 and 256 for the query paper and venue (concatenation of its 100 randomly selected published papers' titles) respectively.

**Author Identification.**

*Problem Definition.* Given a query paper node (with associated text) and a candidate author list (each with his/her published papers), we aim to predict which people in the candidate list is the author of the given query paper.

*Experimental Settings.* We adopt in-batch testing with a testing batch size of 256. We use PREC@1 as the metric. The max sequence length is 32 and 256 for the query paper and author (concatenation of his/her 100 randomly selected published papers' titles) respectively.

**Brand Prediction.**

*Problem Definition.* Given a query item node (with associated text) and a candidate brand list (each with items in the brand), we aim to predict which one in the candidate list is the brand for the given query item.

*Experimental Settings.* We adopt in-batch testing with a testing batch size of 256. We use PREC@1 as the metric. The max sequence length is 32 and 256 for query paper and brand (concatenation of its 100 randomly selected items' texts) respectively.

### A.5.3 LEARN TO SELECT SOURCE RELATIONS

We provide problem definitions and experimental settings for tasks in Section 5.5. The tasks include paper recommendation, paper classification, citation prediction, year prediction, item classification, and price prediction.

**Paper Recommendation.**

*Problem Definition.* Given a query paper node (with associated text) and a candidate paper list (each with associated text), we aim to predict which paper in the candidate list should be recommended to the people who are interested in the query paper.

*Experimental Settings.* We have 1,000 samples in the train set to teach the models how to select source relations, 1,000 samples in the validation set to conduct the early stop, and 100,000 samples in the test set to evaluate the performance of the models. The learning rate is set as 1e-3, the training batch size is 128, and the testing batch size is 256. We conduct the in-batch evaluation with PREC@1 as the metric. All experiments are done on one NVIDIA A6000. We repeat three runs for each model and show the mean and standard deviation in Table 4.

**Paper Classification.**

*Problem Definition.* Given a query paper node (with associated text), we aim to predict what is the category of the paper. The number of paper node categories in academic networks (Geology and Mathematics) is shown in Table 9.

Table 9: Number of paper node categories in academic networks.

| Geology | Mathematics |
|---------|-------------|
| 18 | 17 |

*Experimental Settings.* We have 1,000 samples for each category in the train set to teach the models how to select source relations, 200 samples for each category in the validation set to conduct the early stop, and 200 samples for each category in the test set to evaluate the performance of the models. The learning rate is tuned in 5e-3 and 1e-3, the training batch size is 256, and the testing batch size is 256. We adopt Macro-F1 as the metric. All experiments are done on one NVIDIA A6000. We repeat three runs for each model and show the mean and standard deviation in Table 4.

**Citation Prediction.**

*Problem Definition.* Given a query paper node (with associated text), we aim to predict its future number of citations.

*Experimental Settings.* For both Geology and Mathematics datasets, we extract papers the citation of which is in the range from 0 to 100. We randomly select 10,000 papers from the extracted papers to form the training set, 2,000 papers to form the validation set, and 2,000 papers to form the test set. The learning rate is set as 1e-2 for all compared methods, the training batch size is 256, and the

testing batch size is 256. We adopt RMSE as the metric. All experiments are done on one NVIDIA A6000. We repeat three runs for each model and show the mean and standard deviation in Table 4.

**Year Prediction.**

*Problem Definition.* Given a query paper node (with associated text), we aim to predict the year when it was published.

*Experimental Settings.* For both Geology and Mathematics datasets, we conduct minus operations to make the smallest ground truth year to be 0 (we minus all year numbers by the earliest year, *i.e.,* 1981, in MAG). We randomly select 10,000 papers from the extracted papers to form the training set, 2,000 papers to form the validation set, and 2,000 papers to form the test set. The learning rate is set as 1e-2 for all compared methods, the training batch size is 256, and the testing batch size is 256. We adopt RMSE as the metric. All experiments are done on one NVIDIA A6000. We repeat three runs for each model and show the mean and standard deviation in Table 4.

**Item Classification.**

*Problem Definition.* Given a query item node (with associated text), we aim to predict what is the category of the item. The number of item node categories in e-commerce networks (Clothes, Home, and Sports) is shown in Table 10.

Table 10: Number of item node categories in e-commerce networks.

| Clothes | Home | Sports |
| --- | --- | --- |
| 7 | 9 | 16 |

*Experimental Settings.* We have 1,000 samples for each category in the train set to teach the models how to select source relations, 200 samples for each category in the validation set to conduct the early stop, and 200 samples for each category in the test set to evaluate the performance of the models. The learning rate is tuned in 5e-3 and 1e-3, the training batch size is 256, and the testing batch size is 256. We adopt Macro-F1 as the metric. All experiments are done on one NVIDIA A6000. We repeat three runs for each model and show the mean and standard deviation in Table 5.

**Price Prediction.**

*Problem Definition.* Given a query item node (with associated text), we aim to predict its price.

*Experimental Settings.* For Clothes, Home, and Sports, we delete the long-tail items and keep items whose prices are under 100/1,000/100 respectively. We randomly select 10,000 items from the extracted items to form the training set, 2,000 items to form the validation set, and 2,000 items to form the test set. The learning rate is set as 1e-2 for all compared methods, the training batch size is 256, and the testing batch size is 256. We adopt RMSE as the metric. All experiments are done on one NVIDIA A6000. We repeat three runs for each model and show the mean and standard deviation in Table 5.

### A.6 Relation Embedding Initialization Study.

We study how different relation embedding initialization affects the quality of multiplex representations learned by METERN. We explore three initialization settings: zero vectors initialization, normal distribution initialization, and word embedding initialization. The results of average PREC@1 on different networks are shown in Table 11. From the results, there is no significant difference between the representation learning quality of different initialized relation embeddings.

Table 11: Performance of different relation embedding initialization on different networks.

| Model | Geology | Mathematics | Clothes | Home | Sports |
| --- | --- | --- | --- | --- | --- |
| METERN w/ zero init | 36.31 | **55.26** | 79.99 | 79.68 | 77.59 |
| METERN w/ randn init | **36.43** | 55.20 | **80.19** | 79.69 | 77.56 |
| METERN w/ word init | 36.36 | 55.20 | 80.12 | **79.83** | **77.71** |

A.7   MORE RESULTS ON LEARN TO SELECT SOURCE RELATIONS.

In section 4.2, we propose to let the model learn to select source relations for different downstream tasks and show the learned source relation weight for citation prediction and paper recommendation in Figure 3. In this section, we show more results on the learned source relation weight on academic network downstream tasks in Figure 8 and on e-commerce network downstream tasks in Figure 9. There are four downstream tasks in the academic domain: year prediction, citation prediction, paper classification, and paper recommendation. There are two downstream tasks in the e-commerce domain: price prediction and item classification. From Figure 8 and Figure 9, we can find that different relations can benefit different downstream tasks.

On the year prediction task, the "`co-cited-by`" relation and "`same-author`" relation are more useful. This indicates that papers tend to cite recent papers and authors tend to be active within a short period (*e.g.,* active during the Ph.D. study and stop publishing papers after graduation).

On the citation prediction task, the "`same-author`" relation and "`same-venue`" relation are more beneficial. This implies that the impact of papers is more determined by the published venue and the author who writes them (people tend to follow works from famous researchers).

On the paper classification task, the "`cited-by`" relation is quite useful. This means that papers and their cited papers have a tendency to have the same fine-grained topics.

On the paper recommendation task, the "`cited-by`" relation and "`co-cited-by`" relation dominate. The goal of the paper recommendation task is to recommend similar papers to researchers which may contribute to their own research development. The result is interesting since papers and their cited papers have a tendency to have the same topic, and thus should be recommended to researchers together, and papers in the "`co-cited-by`" relation have already demonstrated that their ideas can be combined and result in a new paper (they are cited by same papers), and thus should be recommended together.

On the price prediction task, the "`co-viewed`" and "`bought-together`" relation matters a lot. This implies that the same user tends to view items and buy together items of a similar price range.

On the item classification task, the "`co-viewed`" relation dominates. This means that items co-viewed by the same user tend to have similar functions.

A.8   RELATION TOKEN EMBEDDING VISUALIZATION.

We visualize the relation token embeddings $Z_{\mathcal{R}}$ learned by METERN with t-SNE (Van der Maaten & Hinton, 2008), projecting those embeddings into a 2-dimensional space. The results on the Geology network and Mathematics network are shown in Figure 10, where the embeddings belonging to the same relation are assigned the same color. From the results, we can find that embeddings belonging to the same relation are close to each other, while those belonging to different relations are discriminative. This demonstrates that METERN can learn to capture the different semantics of different relations by assigning different relations' embeddings to different areas in the latent space.

A.9   MORE RELATED WORK

**Multi-task Learning.**   Multi-task learning refers to learning one model for solving multiple tasks (Crawshaw, 2020). In the natural language processing domain, traditional multi-task learning architectures include feed-forward (Liu et al., 2015), recurrence (Liu et al., 2016), cascaded information (Sanh et al., 2019), and adversarial feature separation (Liu et al., 2017). In more recent years, pretrained language models (Devlin et al., 2019; Liu et al., 2019b) and large language models (Brown et al., 2020) have been demonstrated to be powerful for solving multiple natural language processing tasks. Pretrained language models (Devlin et al., 2019; Liu et al., 2019b) which are trained on a very large corpus across different domains can be fine-tuned and generalized to various tasks. Large language model (Brown et al., 2020) further enhances pretrained language models by scaling up the model to billions or trillions of parameters. In the graph domain, researchers have explored graph neural networks for multi-task prompting Sun et al. (2023).

**Learning on Graphs for Real-World Applications.**   The graph (West et al., 2001) is a data structure that is utilized to describe complex relationships between single units (*e.g.*, user, paper, etc). Learning on graph methods (Hamilton et al., 2017b) are proposed to learn vector representations for nodes based on the graph structure information and node feature information. The learned node

representations can be further utilized in downstream tasks (Hamilton et al., 2017a). In the academic domain, networks (Wang et al., 2020) are constructed with papers as nodes and their relations (*e.g.*, citation, authorship, etc) as edges. The representation learned for papers on such networks can be utilized for paper recommendation (Bai et al., 2019), paper classification (Chowdhury & Schoen, 2020), and author identification (Madigan et al., 2005). In the e-commerce domain (Rajgopal et al., 2003), item networks are adopted to understand different products and promote commercial profit. The product embeddings learned from such networks can be deployed for item recommendation (He et al., 2020), bundle recommendation (Chang et al., 2020), and product understanding (Xu et al., 2019). In the literature domain, we can construct networks with books and authors as nodes, and their relationship as edges. The learned representations for books and authors can be used for book recommendation (Anwar et al., 2019) and author disambiguation (Sanyal et al., 2021). In the social science domain, researchers usually model the interaction between people into a graph structure. In such graphs, the nodes are people and the edges are the interaction between people. The node embeddings learned on such graphs can be adopted for friend recommendation (Chen et al., 2020), user analysis (Wang et al., 2016), and community detection (Shchur & Günnemann, 2019). In the legal domain, opinions given by the judges always contain references to opinions given for previous cases. In such a scenario, people can construct an opinion network (Whalen, 2016) based on the citation relations between opinions. The representations learned on such a network can be utilized for clause classification (Friedrich et al., 2016) and opinion recommendation (Guha et al., 2023).

## A.10 COMPARISON WITH MULTIPLEX GNN

We compare our methods with multiplex GNNs equipped with pretrained text embeddings (MPNet-v2 embedding). The results are shown in Figure 12 and Figure 13.

Table 12: Comparison between multiplex GNNs and METERN on Mathematics network.

| Model | Mathematics | | | | | |
| --- | --- | --- | --- | --- | --- | --- |
| | cb | sa | sv | cr | ccb | Avg. |
| DMGI (BOW) | 46.55 | 42.62 | 6.11 | 27.80 | 38.87 | 28.85 |
| DMGI (MPNet) | 54.13 | 53.06 | 7.40 | 31.39 | 43.98 | 37.99 |
| HDMI (BOW) | 52.65 | 52.71 | 5.54 | 31.80 | 42.54 | 37.05 |
| HDMI (MPNet) | 57.34 | 54.45 | 6.59 | 33.45 | 44.24 | 39.21 |
| Ours | **79.40** | **72.51** | **14.03** | **47.81** | **62.24** | **55.20** |

Table 13: Comparison between multiplex GNNs and METERN on Sports network.

| Model | Sports | | | | |
| --- | --- | --- | --- | --- | --- |
| | cop | cov | bt | cob | Avg. |
| DMGI (BOW) | 41.37 | 46.27 | 41.24 | 31.92 | 40.20 |
| DMGI (MPNet) | 43.37 | 63.69 | 58.16 | 51.72 | 54.24 |
| HDMI (BOW) | 45.43 | 61.22 | 55.56 | 52.66 | 53.72 |
| HDMI (MPNet) | 43.12 | 62.65 | 57.88 | 51.75 | 53.85 |
| Ours | **67.92** | **79.85** | **81.52** | **81.54** | **77.71** |

From the result, we can find that the performance of multiplex GNN with pretrained text embeddings is consistently better than the performance of multiplex GNN with bag-of-word embeddings. However, our method can outperform both baselines by a large margin.

## A.11 COMPARISON WITH NATURAL LANGUAGE DESCRIPTION OF RELATION + LM BASELINES

Another way to capture relation-specific knowledge is to use natural language descriptions of the relations and feed them into the language model for joint relation and text encoding. We implement this baseline by adding a natural language description of the relation before encoding and using the large LM OpenAI-ada-002 (best OpenAI embedding method) as the backbone text encoder. The results on the Mathematics network and the Sports network are shown in Figure 14 and Figure 15.

Table 14: Comparison between relation description baseline and METERN on Mathematics network.

| Model | Mathematics | | | | | |
|---|---|---|---|---|---|---|
| | cb | sa | sv | cr | ccb | Avg. |
| OpenAI-ada-002 (relation description) | 38.87 | 26.40 | 2.43 | 30.72 | 18.74 | 23.43 |
| Ours | **79.40** | **72.51** | **14.03** | **47.81** | **62.24** | **55.20** |

Table 15: Comparison between relation description baseline and METERN on Sports network.

| Model | Sports | | | | |
|---|---|---|---|---|---|
| | cop | cov | bt | cob | Avg. |
| OpenAI-ada-002 (relation description) | 49.56 | 71.75 | 58.36 | 54.63 | 58.58 |
| Ours | **67.92** | **79.85** | **81.52** | **81.54** | **77.71** |

From the result, we can find that our method outperforms this baseline by a large margin, this demonstrates that the relation embeddings learned by our method to represent relations are better than the natural language description used in the baseline.

## A.12 COMPARISON WITH LINKBERT AND GRAPHFORMERS

We compare our method with LinkBERT (Yasunaga et al., 2022) and GraphFormers (Yang et al., 2021) on the Mathematics network and the Sports network. The results are shown in Figure 16 and Figure 17:

Table 16: Comparison between LinkBERT, GraphFormers and METERN on Mathematics network.

| Model | Mathematics | | | | | |
|---|---|---|---|---|---|---|
| | cb | sa | sv | cr | ccb | Avg. |
| LinkBERT | 74.18 | 62.52 | 8.84 | 44.53 | 59.76 | 49.96 |
| GraphFormers | 64.54 | 47.96 | 5.19 | 36.70 | 52.13 | 41.30 |
| Ours | **79.40** | **72.51** | **14.03** | **47.81** | **62.24** | **55.20** |

From the result, we can find that our method outperforms LinkBERT and GraphFormers consistently. The main reason is that these two baseline methods only generate one embedding for each text unit, expecting that all types of relations between texts can be captured by these single-view embeddings, which do not always hold in real-world scenarios. GraphFormers adopts the same graph propagation and aggregation module for edges of different semantics, which results in degraded performance.

Table 17: Comparison between LinkBERT, GraphFormers and METERN on Sports network.

| Model | Sports | | | | |
| --- | --- | --- | --- | --- | --- |
| | cop | cov | bt | cob | Avg. |
| LinkBERT | 67.65 | 76.34 | 80.76 | 77.73 | 75.62 |
| GraphFormers | 66.92 | 78.10 | 79.14 | 78.29 | 75.61 |
| Ours | **67.92** | **79.85** | **81.52** | **81.54** | **77.71** |

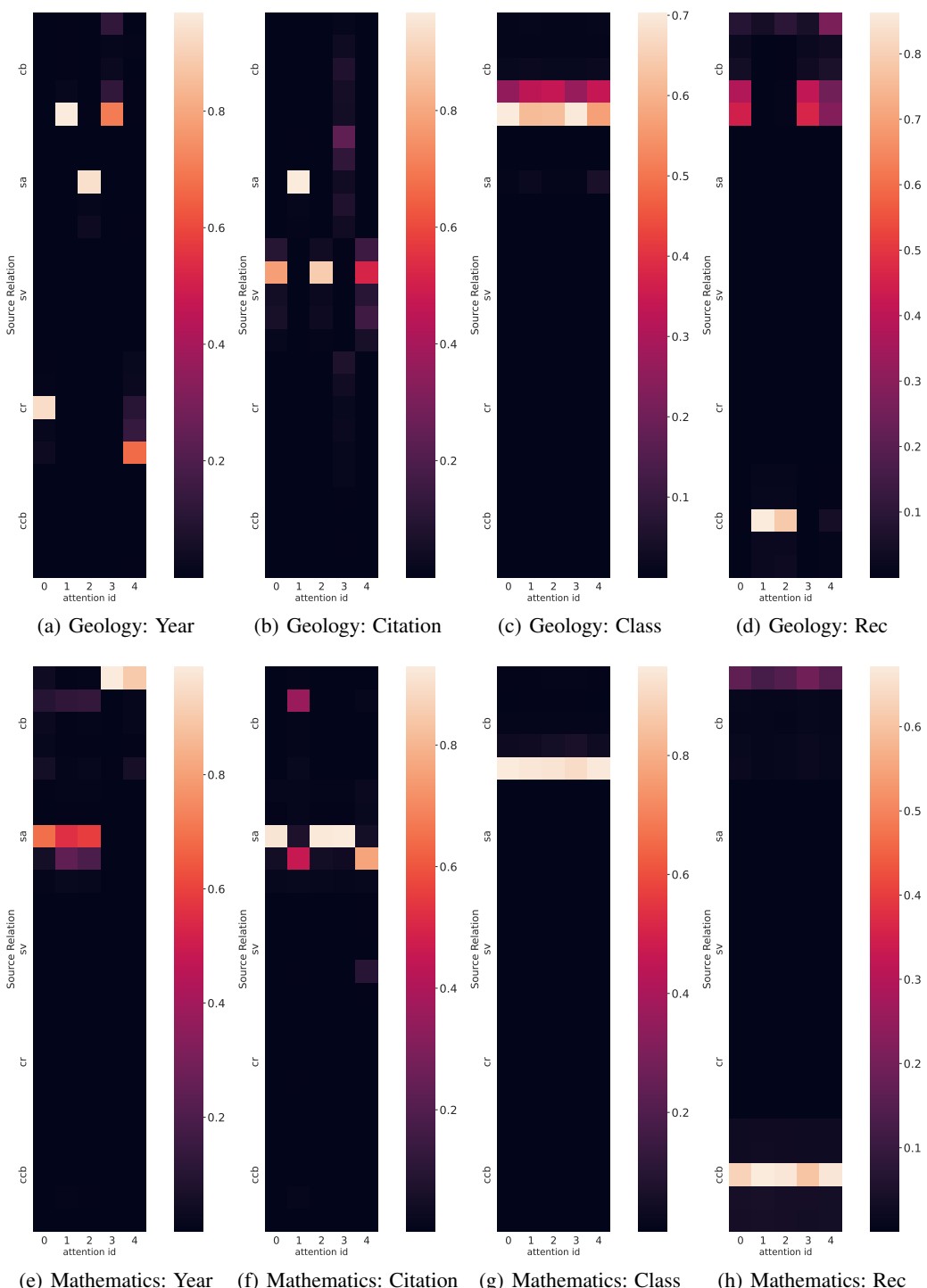

Figure 8: Learnt source relation weights for tasks on academic networks (Geology and Mathematics). The x-axis is the learned weight vector id and the y-axis is the relation embedding id grouped by relation id. cb, sa, sv, cr, and ccb represent "cited-by", "same-author", "same-venue", "co-reference", and "co-cited-by" relation respectively.

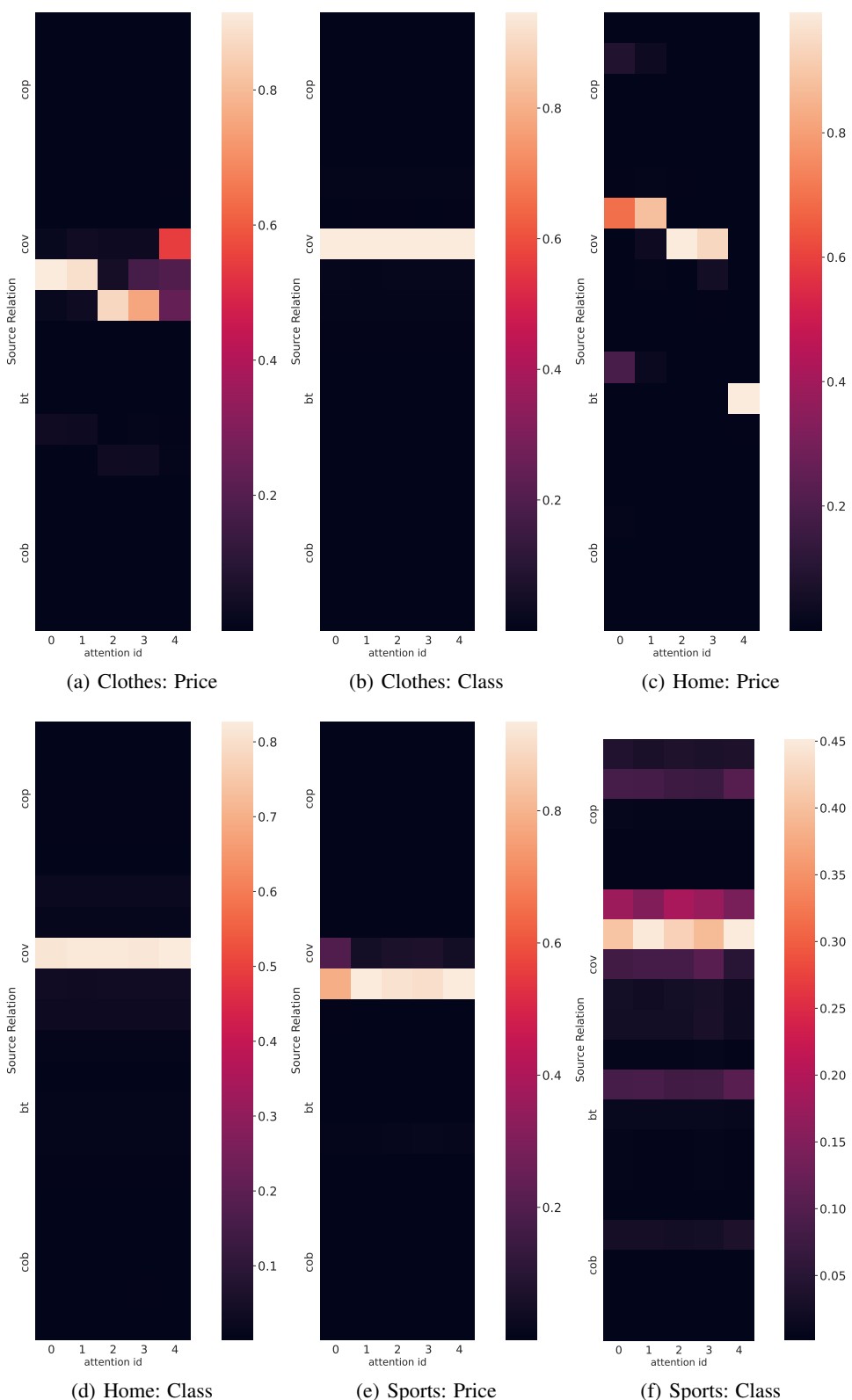

Figure 9: Learnt source relation weights for tasks on e-commerce networks (Clothes, Home, and Sports). The x-axis is the learned weight vector id and the y-axis is the relation embedding id grouped by relation id. cop, cov, bt, and cob represent "`co-purchased`", "`co-viewed`", "`bought-together`", and "`co-brand`" relation respectively.

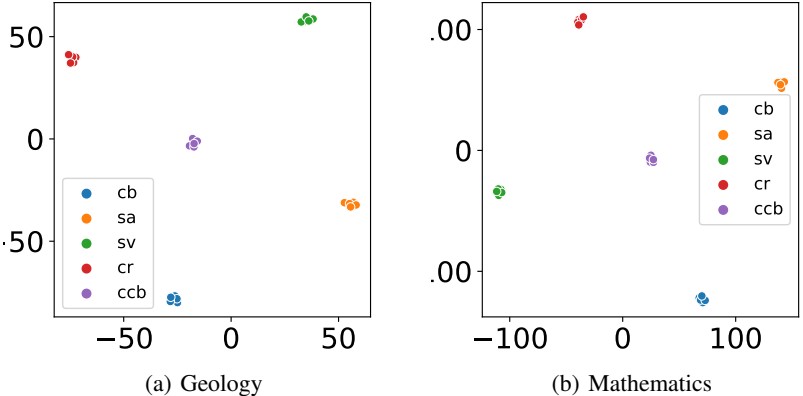

(a) Geology            (b) Mathematics

Figure 10: Visualization of relation embedding $Z_R$ on Geology and Mathematics. cb, sa, sv, cr, and ccb represent "`cited-by`", "`same-author`", "`same-venue`", "`co-reference`", and "`co-cited-by`" relation respectively. We can find that embeddings belonging to the same relation are close to each other, while those belonging to different relations are discriminative.

