# OpenReview forum: "Learning Multiplex Embeddings on Text-rich Networks with One Text Encoder"
_ICLR.cc/2024/Conference — Submitted to ICLR 2024_

### Official Review · Reviewer_NTrJ · 2023-10-29

**Soundness:** 3 good
**Presentation:** 4 excellent
**Contribution:** 3 good
**Rating:** 6
**Confidence:** 4

**Summary:**

This work notices that a real-word text graph usually has multiple types of edges connecting text documents, which is called multiplex network. To account for such multi-type edges, authors propose to assign embeddings for different relations or edge types. The embeddings in turn are concatenated with text tokens for encoding. Furthermore, since some downstream tasks may not have a direct alignment with any observed relation or edge type, authors propose an inference method to learn text embeddings with an attention mechanism for different edge types. Experiments show the effectiveness of the proposed model.

**Strengths:**

1. Clear motivation. This paper clearly introduces the drawback of existing works and explains why a multiplex network is important and why a text should possess multiple embeddings to capture different relations.

2. This paper is self-contained and comprehensive with the evidence that authors present a figure for visual illustration of the model, text description of the modeling component, complexity analysis, as well as a contrast against existing works.

3. Experiments are comprehensive with different tasks and baselines from different categories.

**Weaknesses:**

1. When conducting experiments, we usually encourage authors to repeat the same experiments multiple times and report both mean and standard deviation. However, I see mean results only, but not standard deviation, which is hard for readers to judge the significance of the outperformance over baselines.

2. Insufficient baselines. There are models that work on text-rich networks with pretrained language models, such as GraphFormer [1] and LinkBERT [2]. Though they are designed not for multiplex networks, it is worth comparing against them to highlight the advantage of the multiplex network proposed by this submitted work.

[1] Yang, J., Liu, Z., Xiao, S., Li, C., Lian, D., Agrawal, S., ... & Xie, X. (2021). GraphFormers: GNN-nested transformers for representation learning on textual graph. Advances in Neural Information Processing Systems, 34, 28798-28810.

[2] Yasunaga, M., Leskovec, J., & Liang, P. (2022, May). LinkBERT: Pretraining Language Models with Document Links. In Proceedings of the 60th Annual Meeting of the Association for Computational Linguistics (Volume 1: Long Papers) (pp. 8003-8016).

3. Some math notations are not well defined, such as superscript $m$ at second line in page 4 and superscript $s$ at second last line in page 4. Authors are suggested to provide a table to summarize main mathematical notations used.

**Questions:**

1. How to handle a previously unseen relation during testing phase?
2. what does superscript $m$ mean and how to set a value for it? Same question for superscript $s$.

---

> ### Author Response · Authors · 2023-11-18
> **Response to Reviewer NTrJ**
>
> Thank you so much for your thoughtful review!
>
> Regarding your questions:
> 1. **No standard deviation.** Thank you for the suggestion! We conduct a two-sample independent t-test to compare our method with the best baseline (in each column). Our method outperforms the best baseline method significantly with the p-value of that t-test less than 0.05. We will clarify this in our revision.
>
> 2. **Insufficient baselines.** Thanks for pointing out the two baselines. We agree with the reviewer that it would be interesting to compare these methods, so we add the results on the Mathematics network and Sports network below (also added to Appendix A.12):
>
> Mathematics
>
> | Model                      | cb | sa | sv | cr | ccb | Avg.|
> | -------------------- | ----- | ----- | ---- | ----- | ----- | ----- |
> | LinkBERT                   | 74.18 | 62.52 | 8.84  | 44.53 | 59.76 | 49.96 |
> | GraphFormers               | 64.54 | 47.96 | 5.19  | 36.70 | 52.13 | 41.30 |
> | Ours                       | 79.40 | 72.51 | 14.03 | 47.81 | 62.24 | 55.20 |
>
> Sports
>
> | Model                      | cop | cov | bt | cob | Avg.|
> | ---------------- | ----- | ----- | --- | ----- | ----- |
> | LinkBERT                   | 67.65 | 76.34 | 80.76 | 77.73 | 75.62 |
> | GraphFormers               | 66.92 | 78.10 | 79.14 | 78.29 | 75.61 |
> | Ours                       | 67.92 | 79.85 | 81.52 | 81.54 | 77.71 |
>
>
> From the results, we can find that our method outperforms LinkBERT and GraphFormers consistently. The main reason is that these two baseline methods only generate one embedding for each text unit, expecting that all types of relations between texts can be captured by these single-view embeddings, which do not always hold in real-world scenarios. GraphFormers adopts the same graph propagation and aggregation module for edges of different semantics, which results in degraded performance. That being said, we agree with the reviewer that both GraphFormers and LinkBERT are related studies, we have cited these papers in our revision.
>
> 3. **Math notations: m & s.** Thanks for pointing it out. $m$ refers to the number of relation prior tokens used in relation-conditioned node embedding learning, and $s$ refers to the number of prior tokens used in the downstream task with learning to select. In our experiments, we find the model quite robust to $m$ and $s$. Both $m$ and $s$ are set as 5.
>
> 4. **Handle unseen relations during testing.** We have two ways to conduct inference for an unseen relation during the testing phase: 1) If we do not have any labeled data for the unseen relation, we can use direct inference with an evident source relation (proposed in Sec 4.2). A straightforward way is to select the source relation which has the closest semantics to the unseen relation. 2) If we have some labeled data, we can let METERN learn to select the source relations for the unseen relation inference (proposed in Sec 4.2).

---

### Official Review · Reviewer_WxEq · 2023-11-01

**Soundness:** 3 good
**Presentation:** 3 good
**Contribution:** 3 good
**Rating:** 6
**Confidence:** 3

**Summary:**

The paper introduces a framework for learning multiplex embeddings in text-rich networks. The authors address the challenges of modeling diverse semantic relations among texts in real-world scenarios, where traditional methods based on pre-trained language models (PLMs) fall short due to their single-view embeddings. Additionally, multiplex GNNs cannot capture the semantics of associated texts effectively. METERN overcomes these limitations by using one shared PLM encoder with relation prior tokens to capture both shared knowledge and relation-specific signals. The paper demonstrates the effectiveness of METERN on various downstream tasks across academic and e-commerce networks.

**Strengths:**

- The paper addresses an important and challenging problem in multiplex text-rich networks and proposes an innovative approach, METERN, to tackle it. This approach leverages relation prior tokens and a shared PLM encoder to capture both shared and relation-specific information, providing a unique solution to the problem.

- The authors provide extensive experimental results across a range of downstream tasks and domains, demonstrating the effectiveness of METERN in outperforming competitive baseline methods. This empirical validation adds credibility to the proposed approach and its practical utility.

- The paper discusses the application of METERN in different scenarios, including direct inference and "learn-to-select source-relation" inference, which enhances the versatility of the proposed framework.

**Weaknesses:**

- The paper could benefit from a more detailed and structured discussion of related works. Currently the line spacing is too small and more detailed discussions can provide more context for readers.

- The paper introduces "relation prior tokens" as a novel concept, but lacks thorough discussions of the computational complexity and scalability implications of this approach.

- While the paper emphasizes the problem of multiplex representation learning in text-rich networks, it could benefit from real-world use cases or examples to illustrate the practical applications and impact of the proposed method.

**Questions:**

Can you elaborate on real-world use cases or examples where METERN's multiplex representation learning could have a significant impact, such as in academic networks or e-commerce domains? How might this framework be adopted and adapted in practical scenarios?

---

> ### Author Response · Authors · 2023-11-18
> **Response to Reviewer WxEq**
>
> Thank you so much for your thoughtful review!
>
> Regarding your questions:
>
> 1. **More detailed related works.** We thank the reviewer for the comment. We have added the most relevant paper to our current related work section, but we are happy to add more related works to provide more context for readers. The newly added related works on “multi-task learning” and “learning on graphs for real-world applications” can be found in Appendix A.9. We also appreciate further suggestions for related work.
>
> 2. **Computational complexity.** We conduct the theoretical complexity analysis in Section 4.3 and the empirical complexity analysis in Section 5.6 (Table 6). From both the theoretical analysis and empirical results, we can find that: the time complexity and memory complexity of training METERN are on par with those of training baseline text encoders.
>
> 3. **Real-world use cases and examples.** The METERN model can be deployed to real-world academic paper systems (e.g., Google Scholar) and e-commerce systems (e.g., Amazon). For the academic paper systems, we can use only one model (METERN) to conduct paper analysis from multiple perspectives (semantics, topic, authorship, etc) and achieve paper recommendation, venue recommendation, collaborator recommendation, and paper analysis. For the e-commerce systems, we can deploy METERN to analyze items from different aspects (category, brand, price, etc) and conduct item recommendations, brand recommendations, and item retrieval. Potential real-world scenarios also include the literature domain and legal domain where texts are associated with rich structure information from different aspects. A more detailed discussion of real-world use cases can be found in Section A.9 in the Appendix.

---

> > ### Comment · Reviewer_WxEq · 2023-11-23
> >
> > Dear Authors,
> >
> > Thank you for the response to my comments. The rebuttal has addressed my concerns and I'll keep my score.

---

### Official Review · Reviewer_LwBx · 2023-11-03

**Soundness:** 3 good
**Presentation:** 4 excellent
**Contribution:** 3 good
**Rating:** 6
**Confidence:** 4

**Summary:**

This work focuses on generating multiplex embeddings i.e. relation-conditioned node/text embedding for each relation type. The paper address the challenges of prior art in terms of time and memory, as the the prior art employed one encoder for each relation types in learning multiplex embeddings.
In doing so, they encode multiplex embeddings by introducing prior relation tokens and computing relation-conditioned embeddings with one encoder using the node/text information and the target relation as input such that relation-specific signals are encoded in each relation embeddings and on top, shared knowledge across relations is captured by a single encoder.  The paper also contributes in inference schemes: direct and "learn-to-select-source-relations" inference to map/select source relation and corresponding embeddings for downstream tasks.

Experimental setup is comprehensive, employing several datasets and  baselines approaches. The approach has shown improved performance for downstream tasks and gained efficiency in term of time and memory compared to the baseline approaches.

**Strengths:**

+ the paper is well written and easy to follow
+ motivation, problem statement and methodology is well formulated
+ the paper is incremental however simple in extending the baseline research, like using one encoder for generating relation-conditioned multiplex embeddings instead of employing one encoder for one relation type
+ the paper also contributes in inference strategies for selecting source relations for downstream tasks
+ the proposed approach is incremental, however proposes benefits in terms of compute cost and performance boost on downstream tasks
+ the proposed approach has been evaluated across different data sets, sound exprimental setup and qualitative evaluation
+ sound abalation studies and efficiancy analysis  e.g. time and memory optimization
+ the paper also presents a good study and insights on relation weights and multiplex embedding visualization
+ experimental setup is comprehensive

**Weaknesses:**

- the paper is incremental and advances the baseline approaches in multiplex embedding representation, like introducing a single encoder to generate relation-conditioned multiplex embeddings for each relation type instead of using several encoders
- still there are few questions about claims on scalability (please see question section)
- the scalability of "learning-to-select-source-relations" for downstream tasks for novel relation types is unclear?

**Questions:**

- how does text encoder handle the concept drift induced due to different relation or data distribution or domains?
- how to predetermine the number of relation types and corresponding relation prior tokens?
- In multiplex text-rich network, what is the span (context) of associated texts to compute node representation?

- Transformers in essence are a special case of GNN - they semantically compute a different contextualized representation for each token (or multiple text tokens) in the text sequence and encode different relationship in different contexts. How does it differ with the proposed method of computing multiplex representation for texts for nodes?

- Here, the example are contrary to proposed method - multiplex representation for a token or text sequence for example for the word "llama", however, in the proposed method, node is representing a document, which is longer text and may consists of several thematic structures like topics. Please clarify.

- how does the multiplex representation learning approach scale as the number of relation scales between entities? Similarly, if many relations exists among entities, there are many representations of the same texts due to different relations. How does the approach scale in terms of handling multiplex embedding representations for the sake of leveraging overlapping semantics across relations ?

- On one hand, the method proposes to use only one text encoder to learn different relation embeddings however on other side, generating as many embeddings of the same text in different relations. Scalability is still challenging with this approach in real-world application?

- How does the "learn-to-select-source-relations" inference work for novel relations on the downstream task? the approach seems to works with relations of known type or the downstream task has to select a relation from one of the source relations?

---

> ### Author Response · Authors · 2023-11-18
> **Response to Reviewer LwBx**
>
> Thank you so much for your thoughtful and detailed review!
>
> Regarding your questions:
>
> 1. **Concept drift.** The concept drift does exist across different relations. As shown in Figure 6 in the Appendix, the raw data distributions of text pairs in different relations are quite different. The raw data distribution shift will result in the performance changes shown in Figure 7. In our methods, we use relation embeddings to capture the specific knowledge for different relation distributions and learn the shared knowledge across relations with the text encoder.
>
> 2. **Number of relation types and prior tokens.** The number of *relation types* is predetermined by the network of interest (e.g., for e-commerce item networks, we can have four relation types: co-purchase, co-view, bought-together, and co-brand, according to user behavior and item features). The number of *relation prior tokens* is a hyperparameter in our model. In our experiments, we find the model quite robust to the number of prior tokens and we set it as 5.
>
> 3. **The span of associated texts.** The span of node-associated texts is a flexible choice in multiplex text-rich networks. Our method proposes a generic framework to learn multiplex embeddings and is not limited by the span of text. In our experiments, we use text spans that can be encoded by pretrained language models (less than PLM max length). For papers in academic networks, we use their title/abstract; For items in e-commerce networks, we use their title/descriptions. We can also choose other types of texts (e.g., content for paper) and we leave it for future study.
>
> 4. **Relationship between Transformers and METERN?** This is a great point. We would like to claim the difference between Transformers and METERN from three perspectives: 1) *Node (token v. document).* If we treat Transformers as a special GNN, each node in the graph will be a token and the framework will finally output representations for tokens. However, in METERN, each node corresponds to a document where the model outputs are document representations.  2) *Graph (token complete graph v. real-world semantic graph).* If we treat Transformers as a special GNN, the graph for Transformers will be a complete graph between token nodes, while in our case, the networks are real-world graphs where the edges contain semantic information. 3) *Condition (context v. relation).* If we treat Transformers as a special GNN, the framework will calculate the context-conditioned token (node) representations, while in our framework, we calculate the relation-conditioned document (node) representations.
>
> 5. **The example of “llama”.** Thanks for the comments. The example of “llama” in the Introduction is supposed to claim the weakness of multiplex GNN methods, which represent the texts associated with each node as bag-of-words or context-free embeddings and are not enough to characterize the contextualized text semantics.
>
> 6. **How to handle overlapping semantics across relations.** Thanks for this great comment. 1) During training, the relation prior embeddings will capture the semantics of different relations. If two relations have overlapping semantics, it will be reflected in their relation prior embeddings (similar prior embeddings). 2) During inference, the “learn-to-select” mechanism will select related source relations for the downstream tasks. If two source relations have overlapping semantics, the “learn-to-select” mechanism will automatically learn appropriate weights by accounting for their semantic similarity.
>
> 7. **Scalability of multiplex representation learning.** 1) Training multiplex embeddings: Our method shares similar training time and memory complexity with text encoder baselines (Table 6). 2) Inference for downstream tasks: In practice, we do not need to save the multiplex embeddings for all documents offline. Instead, we only need to save the embeddings of the source relation which are useful for the downstream tasks. In addition, we can save the METERN parameters and conduct online inference (with our proposed two inference mechanisms) for downstream tasks.
>
> 8. **How does “learn-to-select” work for novel relations?** 1) If we have labeled data for this novel source relation, we can retrain the model for multiplex representation learning. Then we will obtain the relation prior embeddings for it for “learn-to-select” inference.  2) If we do not have labeled data for the novel relation but the relation is similar to one or several of our source relations, we can use the prior embeddings corresponding to the similar source relations for this novel relation for “learn-to-select” inference. 3) If we do not have labeled data for the novel relation and the novel relation is fully out of distribution compared with the source relations, there is essentially no signal in the available data to capture the semantics of this new novel relation and it should not be used in “learn-to-select” inference to guarantee performance.

---

### Official Review · Reviewer_SgSL · 2023-11-03

**Soundness:** 2 fair
**Presentation:** 2 fair
**Contribution:** 2 fair
**Rating:** 5
**Confidence:** 3

**Summary:**

The paper introduces an approach to learn embeddings for nodes in multi-relational knowledge graphs where the node embedding is conditioned on the target relation of interest. The relation-informed embedding is to be used for knowledge graph inference.

The approach to learn the embedding is to include prefix embeddings to the language model encoder  that are relation-specific similar to prefix tuning. The relation embeddings are learned from the training data by maximizing similarity of observed node pairs as is typical for knowledge graph embedding. At inference time, when it is not apparent what relevant relation embedding to use , the authors propose an attention over the trained relation embeddings that is learned from some limited samples from the target task.

The authors perform evaluation of their approach, comparing with some existing approaches (some of which like GNN do not use language model encoders) and some single node embedding baselines. The approach seems somewhat better than the baselines.

**Strengths:**

- The paper seems easy to follow and the problems is clearly described.
- Results show good improvements over baselines.

**Weaknesses:**

- The approach is very much prefix tuning applied to this problem of knowledge graph inference, so as such there is little novelty in the approach.
- Baselines:
   - some of the baselines like multiplex GNN use weaker bag of word encoder so they are not directly comparable.
   - other baselines it's hard to see which baselines actually use a comparable encoder to the proposed model, ideally all baselines should have the same underlying encoder
- The approach is missing comparison with the simpler, no training required, baseline of adding a natural language description of the relation before encoding (instead of learned prefix embeddings). This approach is also directly applicable at inference time without requiring any additional training for unseen relations. This should potentially be applied with larger LM embeddings as well since there is no additional training needed.

**Questions:**

See "weakness" section for concerns with baselines.

---

> ### Author Response · Authors · 2023-11-18
> **Response to Reviewer SgSL**
>
> Thank you so much for your thoughtful review!
>
> Regarding your questions:
> 1. **Novelty**. *Focus*. We would like to argue that existing works [1] mainly focus on efficient model training, which is different from the focus of our work, which is to explore a new task of learning relation-conditioned document representations (i.e., multiplex representations on text-rich networks). *Techniques.* We propose a simple and effective multi-task learning method tailored for the multiplex nature of text-rich networks. The learned relation prior tokens capture the semantics of relations. These tokens can be flexibly used in the inference process either through manual selection, facilitating a plug-and-play functionality, or serving as prototype representations upon which mixup weights are automatically learned to quickly adapt to potentially intricate and unseen relations. *Applications.* The learned multiplex representations not only benefit relation prediction (link prediction corresponding to a specific relation) but also help various downstream tasks (shown in Tables 3,4, and 5) on the text-rich networks. We propose METERN to unify all task-solving with one model on multiplex text-rich networks.
>
> 2. **GNN baselines.** Thanks for pointing out the two baselines. We agree that it would be interesting to compare with a multiplex GNN equipped with pretrained text embeddings (MPNet-v2 is used here. We also tried BERT, and found that MPNet-v2 has better performance). We show the results on the Mathematics network and Sports network below (also added to Appendix A.10):
>
> Mathematics
>
> | Model | cb | sa | sv | cr | ccb| Avg.  |
> |--------|-------|-------|-------|-------|-------|-------|
> | DMGI (BOW)              | 46.55 | 42.62 | 6.11  | 27.80 | 38.87 | 28.85 |
> | HDMI (BOW)              | 52.65 | 52.71 | 5.54  | 31.80 | 42.54 | 37.05 |
> | DMGI (MPNet)            | 54.13 | 53.06 | 7.40  | 31.39 | 43.98 | 37.99 |
> | HDMI (MPNet)            | 57.34 | 54.45 | 6.59  | 33.45 | 44.24 | 39.21 |
> | Ours                    | 79.40 | 72.51 | 14.03 | 47.81 | 62.24 | 55.20 |
>
> Sports
>
> | Model| cop | cov | bt | cob | Avg.|
> |---------|-----|-----|----|-----|-----|
> | DMGI (BOW)             | 41.37 | 46.27 | 41.24 | 31.92 | 40.20 |
> | HDMI (BOW)             | 45.43 | 61.22 | 55.56 | 52.66 | 53.72 |
> | DMGI (MPNet)           | 43.37 | 63.69 | 58.16 | 51.72 | 54.24 |
> | HDMI (MPNet)           | 43.12 | 62.65 | 57.88 | 51.75 | 53.85 |
> | Ours                   | 67.92 | 79.85 | 81.52 | 81.54 | 77.71 |
>
> From the results, we can find that the performance of multiplex GNNs with pretrained text embeddings is consistently better than the performance of multiplex GNNs with bag-of-words embeddings. However, our method still outperforms both baselines by large margins.
>
> 3. **Text encoder.** We compare METERN with three types of baselines: 1) large-scale corpora finetuned text encoders, 2) multiplex graph neural networks, and 3) multi-relation learning language models. They have different philosophies to capture the multiplex relation semantics. The first category of methods is finetuned on very large text corpora, generating single embeddings for documents, and presuming one single embedding can capture the multiplex relation and have good generalization ability. The second category of methods learns multiplex node representations from a graph structure perspective. We show the result of this category with pretrained text embeddings in our previous answer. The third category of methods tries to capture different relation semantics with one text encoder and all these methods (including ours) adopt BERT-base as the initial text encoder for a fair comparison.
>
> 4. **Natural language description of relation + LM baseline.** Thanks for raising this baseline. We implement this baseline by adding a natural language description of the relation before encoding and using the large LM OpenAI-ada-002 (the best OpenAI embedding method) as the backbone text encoder. The results are shown below (added to Appendix A.11):
>
> Mathematics
> | Model  | cb | sa | sv | cr | ccb | Avg.|
> | --------- | ----- | ----- | ---- | ----- | ----- | ----- |
> | OpenAI-ada-002 (relation description) | 38.87 | 26.40 | 2.43  | 30.72 | 18.74 | 23.43 |
> | Ours                    | 79.40 | 72.51 | 14.03 | 47.81 | 62.24 | 55.20 |
>
> Sports
> | Model | cop | cov | bt | cob | Avg.|
> | ----------- | ----- | ----- | ---- | ----- | ----- |
> | OpenAI-ada-002 (relation description)| 49.56 | 71.75 | 58.36 | 54.63 | 58.58 |
> | Ours                   | 67.92 | 79.85 | 81.52 | 81.54 | 77.71 |
>
> From the results, we can find that our method outperforms this baseline by a large margin, which demonstrates that the relation embeddings learned by our method to represent relations are better than directly prompting large LMs via natural language descriptions.
>
> [1] Prefix-Tuning: Optimizing Continuous Prompts for Generation. ACL 2021.

---

### Public Comment · ~Juri_Opitz2 · 2023-11-12
**Interesting paper, pointer for related work**

Hi,

your paper has a really interesting topic, I agree that coarsely merging all semantic aspects into one embedding in many situations is not ideal!

In the RW on text embeddings, the paper says "All the existing works presume that one embedding can capture the general semantics for each text unit and do not take the diverse text relation semantics into consideration."

Maybe it might be interesting to consider our work, where we provide a way for partitioning sentence embeddings (e.g., SBERT) into multiple sub-embeddings that capture different semantic relations [1]. Maybe using our model (pre-trained checkpoints are available) or tuning the model with your aspects can be an interesting baseline.

[1] https://arxiv.org/abs/2206.07023

---

> ### Author Response · Authors · 2023-11-18
>
> Thank you so much for pointing out this interesting paper! I think the idea of partitioning sentence embeddings into multiple sub-embeddings is great. We will definitely look into it and we believe we can take inspiration from your work.

---

### Author Response · Authors · 2023-11-18
**General Response**

Dear Reviewers,

We sincerely appreciate your valuable feedback and suggestions. We have revised our paper based on your reviews. We highlighted changes in the manuscript using blue color.

We also want to thank the Reviewers for noting the strengths of our paper, namely:

- The problem addressed in our paper is important and well-motivated. (LwBx, WxEq, NTrJ)
- Our proposed method is substantial and sound. (LwBx, WxEq)
- The paper is clearly written. (SgSL, LwBx, NTrJ)
- The empirical results are consistent, comprehensive, and convincing. (SgSL, LwBx, WxEq, NTrJ)

We have addressed the individual questions of reviewers in separate responses. In the revised version, we incorporated all reviewers' suggestions by adding more clarifications of our method, more experimental results and baselines, as well as more related works. Here we briefly outline the updates to the revised submission for the reference of reviewers.

- [Appendix A.9] We add a more detailed related work section as pointed out by WxEq.
- [Appendix A.10] Some experimental results and analysis for multiplex GNN with pretrained text embeddings. (SgSL)
- [Appendix A.11] The implementation and results of a relation description + LM baseline. (SgSL)
- [Appendix A.12] Some experimental results and analysis for LinkBERT and GraphFormers. (NTrJ)

In closing, we thank the reviewers again for their time and valuable feedback. If you have further concerns, please let us know, and we will be happy to address them.

---

### Author Response · Authors · 2023-11-22
**Kind Reminder**

Dear reviewers,

We want to sincerely thank you again for your involvement and thoughtful feedback! We hope our response addresses your questions.
We would like to gently remind you that we are approaching the end of the discussion with author involvement. In light of the newly added baselines (reviewer SgSL, NTrJ), response to questions of our model (review LwBx), and newly added related work/real-world cases section (reviewer WxEq), we are kindly asking if you are willing to give a higher assessment to our submission. If you have any more thoughts, we are happy to continue our discussion until the deadline.

Thank you so much!

---

### Author Response · Authors · 2023-11-23
**Kind reminder**

We wish to express our sincere gratitude once again to the reviewers for their valuable contributions and considerate feedback. We would like to gently bring to the reviewers' attention that the interaction phase between authors and reviewers is nearing completion (within 12 hours).

Given the inclusion of the new experiments, further explanation of our methods, and a more detailed related work section, we kindly inquire whether the reviewers might consider assigning a more favorable evaluation to our submission. Should you have any further insights to share, we are more than willing to sustain our discussion until the deadline.

---

### Meta-Review · Area_Chair_hB43 · 2023-12-22

**Metareview:**

The paper proposes a model to learn representations for multiplex network, which they define as a graph with documents associated with nodes and multiple attributed relation among nodes. The network is used to be known as heterogenous information network. The authors may want to clarify what is the new for multiplex network, or reuse the classical term if they are the same.

The key idea of this approach is to learn multiple embeddings for each relation, and adding those to the document text embedding for each node. The paper also propose a method to select the relation embedding at inference. Experiments show the effectiveness of the proposed model.

Strengths of the paper:
1. The idea and method are novel. I find it an interesting approach to solve a prediction problem on heterogenous networks.
2. it is parameter efficient. Comparing to the text encoder model, it only introduces a few additional embeddings for each relation. For a small set of relations, it does not introduce many parameters.

Weakness of the paper:
1. The experimental setup does not represent the real setting for academic and e-commerce scenarios. Reviewer WxEq also raised this question but the author was just repeating the statement in the paper. The key issue is the academic and e-commerce graphs are not the original graphs but derived ones. The original academic graph contains papers and authors as nodes, and there are links between authors and papers, and  citations among papers.  While the graph used in this paper is a co-author/co-cite graph which is derived from the original one. If there are co-author links between two papers, it does not tell who is the co-author.
Similarly the e-commerce graph is a derived graph, rather than the original graph between user-item-seller. Why does it matter? Because on the derived graph, the prediction of co-author does not tell who is the author. The co-view would make some sense but not too much because this is not the way used in recommender system.
2. Comparison with multiplex GNN is missing (as raised by Reviewer SgSL). The authors provide results for MPNet, but miss MuxGNN (Melton et al TPAMI 2023) and MHGCN (Yu et al, KDD 2022) which are directly on multiplex heterogenous graphs.
Unfortunately, there were not adequate interaction between reviewers and authors to clarify and justify.
Based on existing information, it is really a borderline decision.

**Justification For Why Not Higher Score:**

This is really a borderline paper. The experiment setup is unrealistic. But I am willing to adjust and accept the paper.

**Justification For Why Not Lower Score:**

N/A

---

### Decision · Program_Chairs · 2024-01-16

Reject